# Malaria parasites require a divergent heme oxygenase for apicoplast gene expression and biogenesis

Amanda Mixon Blackwell[1], Yasaman Jami-Alahmadi[2], Armiyaw S Nasamu[3], Shota Kudo[4], Akinobu Senoo[5], Celine Slam[1], Kouhei Tsumoto[4,6], James A Wohlschlegel[2], Jose Manuel Martinez Caaveiro[4], Daniel E Goldberg[3]*, Paul A Sigala[1,3]*

[1]Department of Biochemistry, University of Utah School of Medicine, Salt Lake City, United States; [2]Department of Biological Chemistry, University of California, Los Angeles, Los Angeles, United States; [3]Departments of Medicine and Molecular Microbiology, Washington University School of Medicine, St. Louis, United States; [4]Department of Chemistry & Biotechnology, The University of Tokyo, Tokyo, Japan; [5]Department of Protein Drug Discovery, Graduate School of Pharmaceutical Sciences, Kyushu University, Fukuoka, Japan; [6]Department of Bioengineering, University of Tokyo, Tokyo, Japan

*For correspondence:
dgoldberg@wustl.edu (DEG);
p.sigala@biochem.utah.edu
(PAS)

Competing interest: The authors declare that no competing interests exist.

## eLife Assessment

This study reveals that the malaria parasite protein PfHO, although lacking typical heme oxygenase activity, is essential for the survival of *Plasmodium falciparum*. Structural and localization analyses demonstrated that PfHO plays a critical role in maintaining the apicoplast, specifically in gene expression and biogenesis, suggesting an adaptive function for this protein in parasite biology. While the findings **convincingly** support the authors' claims, further investigation into apicoplast gene expression and the specific function of PfHO remains a future challenge. The topic and results are **important** and will be of interest to researchers studying various aspects of malaria, Plasmodium physiology, host-pathogen interactions, and heme metabolism.

**Abstract** Malaria parasites have evolved unusual metabolic adaptations that specialize them for growth within heme-rich human erythrocytes. During blood-stage infection, *Plasmodium falciparum* parasites internalize and digest abundant host hemoglobin within the digestive vacuole. This massive catabolic process generates copious free heme, most of which is biomineralized into inert hemozoin. Parasites also express a divergent heme oxygenase (HO)-like protein (PfHO) that lacks key active-site residues and has lost canonical HO activity. The cellular role of this unusual protein that underpins its retention by parasites has been unknown. To unravel PfHO function, we first determined a 2.8 Å-resolution X-ray structure that revealed a highly α-helical fold indicative of distant HO homology. Localization studies unveiled PfHO targeting to the apicoplast organelle, where it is imported and undergoes N-terminal processing but retains most of the electropositive transit peptide. We observed that conditional knockdown of PfHO was lethal to parasites, which died from defective apicoplast biogenesis and impaired isoprenoid-precursor synthesis. Complementation and molecular-interaction studies revealed an essential role for the electropositive N-terminus of PfHO, which selectively associates with the apicoplast genome and enzymes involved in nucleic acid metabolism and gene expression. PfHO knockdown resulted in a specific deficiency in levels of apicoplast-encoded RNA but not DNA. These studies reveal an essential function for PfHO in

apicoplast maintenance and suggest that *Plasmodium* repurposed the conserved HO scaffold from its canonical heme-degrading function in the ancestral chloroplast to fulfill a critical adaptive role in organelle gene expression.

## Introduction

Malaria remains a devastating infectious disease marked by increasing treatment failures with frontline artemisinin therapies (*Naß and Efferth, 2019*; *Lubell et al., 2014*). *Plasmodium* malaria parasites diverged early in eukaryotic evolution from well-studied yeast and mammalian cells, upon which most understanding of eukaryotic biology is based. Due to this evolutionary divergence, parasites acquired unusual molecular adaptations for specialized growth and survival within human erythrocytes, human hepatocytes, and mosquitos. Unraveling and understanding these adaptations will provide deep insights into the evolution of *Plasmodium* and other apicomplexan parasites and unveil new parasite-specific vulnerabilities that are suitable for therapeutic targeting to combat increasing parasite drug resistance.

Heme metabolism is central to parasite survival within red blood cells (RBCs), the most heme-rich cell in the human body. During blood-stage growth, *Plasmodium* parasites internalize and digest up to 80% of host hemoglobin within the acidic digestive vacuole (DV; *Ball et al., 1948*; *Francis et al., 1997*). This massive digestive process liberates an excess of cytotoxic free heme that is detoxified in situ within the DV via biomineralization into chemically inert hemozoin crystals. Other hematophagous organisms and cells, including certain blood-feeding insects and human liver/splenic macrophages that process senescent RBCs, depend on canonical heme oxygenase (HO) enzymes to degrade excess heme and recycle iron (*Elbirt and Bonkovsky, 1999*; *Vijayan et al., 2018*; *Bottino-Rojas et al., 2019*). In contrast to these examples, *Plasmodium* parasites lack an active HO pathway for enzymatic heme degradation and rely fully on alternative mechanisms for heme detoxification and iron acquisition during blood-stage infection (*Sigala et al., 2012*). Nevertheless, malaria parasites express a divergent HO-like protein (PfHO, Pf3D7_1011900) with unusual biochemical features (*Okada, 2009*). Although this protein shows distant homology to HO enzymes, it lacks the strictly conserved active-site His ligand and does not degrade heme in vitro or in live cells (*Sigala et al., 2012*). Genome-wide knockout and insertional mutagenesis studies in *P. berghei* and *P. falciparum* reported that the PfHO gene was refractory to disruption (*Bushell et al., 2017*; *Zhang et al., 2018*), but the biological function that underpins the retention and putative essentiality of this divergent HO-like protein in malaria parasites has remained a mystery.

Heme oxygenases are ubiquitous enzymes that retain a conserved α-helical structure and canonically function in regioselective cleavage of the porphyrin macrocycle of heme to release iron (*Wilks, 2002*; *Tenhunen et al., 1968*). HO-catalyzed heme degradation also generates carbon monoxide and a tetrapyrrole cleavage product, typically biliverdin IXα, which is further modified for downstream metabolic utilization or excretion. These reactions play key roles in heme turnover, iron acquisition, oxidative protection, and cellular signaling (*Elbirt and Bonkovsky, 1999*; *Vijayan et al., 2018*; *Davis et al., 2001*). There are also reports of expanded biological roles for HO proteins that are independent of heme degradation. In humans, HO1 has been reported to translocate to the nucleus upon proteolytic processing where it modulates the activity of transcription factors through an unknown mechanism that is independent of its enzymatic activity (*Lin et al., 2007*). HO1-mediated transcriptional changes have been implicated in cell differentiation and physiological stress responses in humans and rats (*Mascaró et al., 2021*; *Jagadeesh et al., 2022*). Pseudo-HO enzymes that retain the HO fold but lack the conserved His ligand or heme-degrading function have also been identified in multiple organisms, but their cellular roles remain largely unknown. The best-studied example is from *Arabidopsis thaliana*, which encodes three active (HY1, HO3, and HO4) and one inactive (HO2) heme oxygenase homologs that all localize to the chloroplast (*Gisk et al., 2010*). AtHO2 contains an Arg in place of the conserved His and lacks detectable HO activity, and knockout studies suggest an undefined role in photomorphogenesis (*Davis et al., 2001*). Based on these reports of non-canonical HO roles, we set out to unravel the cellular function of PfHO in *P. falciparum*.

We localized PfHO to the parasite apicoplast and demonstrated that conditional knockdown of PfHO disrupts parasite growth and apicoplast biogenesis. We discovered that PfHO interacts with the 35 kb apicoplast genome and requires the electropositive N-terminus, which serves as an

apicoplast-targeting transit peptide but is largely retained after organelle import. Loss of PfHO resulted in a strong deficiency in apicoplast-encoded RNA but not DNA levels, suggesting a key role in expression of the apicoplast genome. Phylogenetic analyses indicated that PfHO orthologs are selectively retained by hematozoan parasites, including *Babesia* and *Theileria*, but not by other apicomplexan organisms. This study illuminates an essential role for a catalytically inactive HO-like protein in gene expression and biogenesis of the *Plasmodium* apicoplast.

## Results

### PfHO is a divergent HO homolog

Sequence analysis of PfHO provides limited insights into the origin and function of this protein. Previous studies identified low (15–20%) sequence identity between PfHO and known heme oxygenases from humans (human HO1, HuHO1), cyanobacteria (*Synechocystis* sp. PCC 6803 HO1, SynHO1), and plants (*A. thaliana* HO4, AtHO4; *Sigala et al., 2012*; *Okada, 2009*). The N-terminal 95 residues of PfHO, which show sequence similarity only to other *Plasmodium* orthologs, display hydrophobic and electropositive features suggestive of sub-cellular targeting (*Figure 1A*), which is discussed below. Sequence homology searches with PfHO using NCBI BLAST (*Altschul et al., 1990*) and Hidden Markov Modeling tools (*Potter et al., 2018*) revealed highest identity (65–99%) to protein orthologs in other *Plasmodium* species, with the next highest sequence identity (30–35%) to proteins in the hematozoan parasites, *Babesia* and *Theileria* (*Figure 1—figure supplement 1*). These PfHO orthologs also lack the conserved active-site His residue and other sequence features required for enzymatic function. However, we were unable to identify convincing PfHO orthologs in apicomplexan organisms outside of blood-infecting hematozoan parasites, as previously reported (*Kloehn et al., 2021*). Phylogenic analysis of HO sequences from mammals, plants, algae, insects, and parasites further highlighted the divergence of heme oxygenases between these clades of organisms (*Figure 1—figure supplement 2*).

The low level of sequence identity to known HO enzymes and loss of heme-degrading activity by PfHO strongly suggested a repurposing of the HO scaffold for an alternative function in *Plasmodium*. As an initial step towards understanding this divergent function, we determined a 2.8 Å-resolution X-ray crystal structure of the HO-like domain of PfHO (residues 84–305) in its unliganded state (*Figure 1B* and *Figure 1—figure supplement 3*), as we were unable to crystalize PfHO$^{84-305}$ in the presence of bound heme. The structure obtained for apo-PfHO$^{84-305}$ was highly α-helical with an overall fold expected for an HO homolog (*Figure 1B* and *Figure 1C*), including strong concordance in the positions of α-helix-forming sequences between PfHO, HuHO1, and SynHO1 (*Figure 1A*). Although the purified protein contained residues 84–305, electron density to support structural modeling began with residue 95 at the start of an α-helix, suggesting that residues 84–94 are disordered. A structural homology search using the DALI Protein Structure Comparison Server (*Holm et al., 2023*) revealed strong structural similarity between PfHO and structures of HO enzymes from plants, mammals, and cyanobacteria (*Figure 1D*). Superposition of crystal structures for PfHO and SynHO1 revealed very similar α-helical folds, with a root-mean-square deviation (RMSD) in the positions of backbone atoms of 2.11 Å for the two structures (*Figure 1B*).

Despite overall structural similarity between PfHO and SynHO1, we noted several points of structural divergence beyond loss of the conserved His ligand. In structures of HO enzymes (including SynHO1), heme is sequestered within an active-site binding pocket formed by a distal helix positioned above the bound heme and a proximal helix below the heme that contains the conserved His ligand (*Figure 1B*). In our PfHO structure, the distal helix adopted a similar architecture to that observed in the SynHO1 structure but featured bulkier, charged Lys-Glu residues in place of the conserved Gly-Gly sequence in active HOs (*Lad et al., 2005*; *Figure 1A*). In contrast to the ordered distal helix, we were unable to resolve the structure of the C-terminal region of the proximal helix (residues 112–133) of PfHO due to weak electron density for these residues, possibly reflecting static or dynamic structural disorder in this region. We note that ordered, coiled loops were observed at the C-terminal end of the proximal helix in recent structures of plant HOs (*Wang et al., 2022b*; *Tohda et al., 2021*; *Figure 1—figure supplement 4*). Nevertheless, disorder in the C-terminal region of the proximal helix has previously been described in a structure of human HO1 bound to synthetic 5-phenylheme

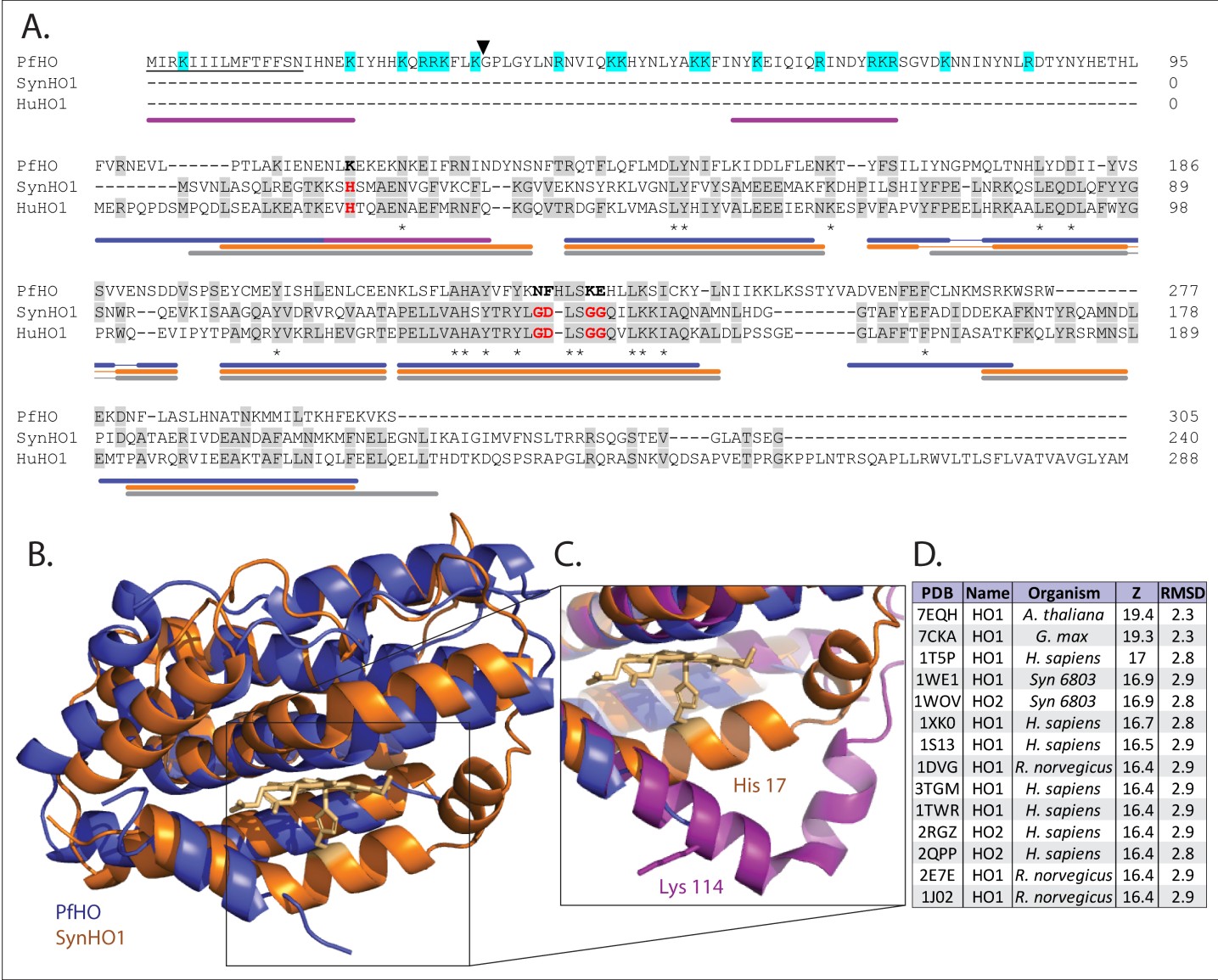

**Figure 1.** Sequence and structural homology of PfHO. (**A**) Sequence alignment of *P. falciparum* (PfHO, Q8IJS6), cyanobacterial (SynHO1, P72849), and human (HuHO1, P09601) heme oxygenase homologs (Uniprot ID). Conserved histidine ligand and distal helix residues required for catalysis in SynHO1 and HuHO1 are marked in red, and identical residues in aligned sequences are in gray. Asterisks indicate identical residues in all three sequences. The predicted N-terminal signal peptide of PfHO is underlined, electropositive residues in the PfHO leader sequence are highlighted in cyan, and the black arrow marks the putative targeting peptide processing site. Colored bars below the sequence alignment mark locations of α helices observed in crystal structures of PfHO (blue), SynHO1 (orange), and HuHO1 (grey), and the AlphaFold structural prediction for PfHO (purple). (**B**) Structural superposition of the 2.8 Å-resolution X-ray crystal structure of apo-PfHO84-305 (blue, PDB: 8ZLD) and the 2.5 Å-resolution X-ray structure of cyanobacterial, SynHO1 (orange, PDB: 1WE1). (**C**) Structural superposition of the proximal helix for SynHO1 active site (orange), PfHO crystal structure (blue), and the AlphaFold structural prediction of PfHO (purple). (**D**) Top-scoring protein structures in the PDB identified by the DALI server based on structural similarity to the X-ray crystal structure of PfHO84-305. RMSD is calculated in angstroms (Å), and Z-score is a unitless parameter describing similarity, where greater value indicates higher similarity (*Holm, 2020*).

The online version of this article includes the following source data and figure supplement(s) for figure 1:

**Source data 1.** PDB file for 2.8 Å-resolution structure of PfHO.

**Figure supplement 1.** Sequence homology of PfHO.

**Figure supplement 2.** Phylogenic tree of mammalian, plant, algal, and hematozoan HOs.

**Figure supplement 3.** X-ray crystallographic data collection and structure refinement statistics for PfHO.

**Figure supplement 3—source data 1.**

*Figure 1 continued on next page*

*Figure 1 continued*

**Figure supplement 4.** Sequence and structural alignment of PfHO and plant HOs.

**Figure supplement 5.** HO surface charge features.

(where the 5-phenyl substituent sterically disrupts proximal helix structure near the α-meso carbon) and a structure of apo rat HO1 that lacked bound heme (*Sugishima et al., 2002*; ; *Wang et al., 2004*).

To model a possible structure for the proximal helix of PfHO as a basis for evaluating its electrostatic properties, we turned to a predicted AlphaFold (*Jumper et al., 2021*) structure for PfHO, which was very similar to our crystal structure (backbone RMSD of 0.57 Å). AlphaFold predicts a sharp kink in the proximal helix of PfHO that extends the position of this helix by several angstroms compared to the proximal helix in SynHO1 (*Figure 1C*). We also noted that the PfHO AlphaFold model predicted that residues 84–94 were unstructured, consistent with our inability to observe electron density for these residues in our X-ray data. We identified changes in the calculated electrostatic surface potential (*Baker et al., 2001*) of PfHO$^{84-305}$ between the proximal and distal helices that diminish the electropositive potential in this region compared to HuHO1 and SynHO1 (*Figure 1—figure supplement 5*). The positive charge character around the heme-binding pocket in canonical HOs interacts electrostatically with the propionate groups of heme and mediates HO association with electronegative electron donors (e.g. ferredoxin and cytochrome P450 reductase) required for HO activity (*Sugishima et al., 2002*; *Sugishima et al., 2004*; *Wang and de Montellano, 2003*). Based on these observations, we conclude that PfHO is a divergent HO homolog that retains the overall HO fold but has lost key active-site and surface features that suggest a unique function independent of heme degradation.

## PfHO is targeted to the apicoplast organelle

The *Plasmodium*-specific N-terminus of PfHO has sequence features that suggested a possible role in sub-cellular targeting. These features include a hydrophobic stretch of ~12 residues at the N-terminus followed by electropositive sequence of ~80 residues that are characteristic of signal and transit peptides, respectively, which direct proteins to the apicoplast organelle (*Waller et al., 2000*; *Zuegge et al., 2001*). Analysis using the apicoplast-targeting prediction software, PlasmoAP, identified strong sequence characteristics of an apicoplast transit peptide, but its SignalP 2.0 module failed to identify a signal peptide (*Sigala et al., 2012*; *Zuegge et al., 2001*). However, more recent SignalP versions (5.0 and 6.0; *Almagro Armenteros et al., 2019*) and Phobius (*Käll et al., 2004*) strongly predicted a signal peptide with a consensus cleavage site after N18 (*Figure 1A*). These features, together with known HO targeting to chloroplasts in plants (*Gisk et al., 2010*) and prior detection of PfHO in pull-down studies of apicoplast-targeted proteins (*Mallari et al., 2014*), suggested that PfHO likely targets to the apicoplast.

To determine PfHO localization in *P. falciparum* parasites, we stably transfected Dd2 parasites with an episome encoding full-length PfHO with a C-terminal GFP-tag. Live parasite microscopy revealed focal GFP signal that was proximal to but distinct from MitoTracker Red staining of the mitochondrion (*Figure 2A* and *Figure 2—figure supplement 1*), which is consistent with PfHO targeting to the apicoplast. Immunofluorescence analysis (IFA) of fixed parasites indicated strong co-localization between PfHO-GFP and the apicoplast acyl carrier protein (ACP, Pf3D7_0208500; *Figure 2A* and *Figure 2—figure supplement 2*), providing direct evidence that PfHO targets the apicoplast. Additionally, western blot analysis of parasite lysates revealed two bands by anti-GFP staining that are suggestive of precursor and N-terminally processed forms, as typically observed for apicoplast-targeted proteins (*Waller et al., 2000*; *Figure 2B*).

To further test this conclusion, we stably disrupted the apicoplast by culturing parasites for 1 wk in 1 µM doxycycline (Dox) with rescue by 200 µM isopentenyl pyrophosphate (IPP) to decouple parasite viability from apicoplast function. In these conditions, the apicoplast is lost and proteins targeted to the organelle accumulate in dispersed cytoplasmic foci (*Yeh and DeRisi, 2011*). As expected for an apicoplast-targeted protein, episomally expressed PfHO-GFP in Dox/IPP-treated parasites displayed a speckled constellation of dispersed fluorescent foci in each cell (*Figure 2C* and *Figure 2—figure supplement 1*). Furthermore, western blot analysis of parasite lysates after Dox/IPP-treatment revealed only a single band by anti-GFP staining at the size of the precursor protein, providing evidence that PfHO processing depends on import into the apicoplast (*Figure 2B*).

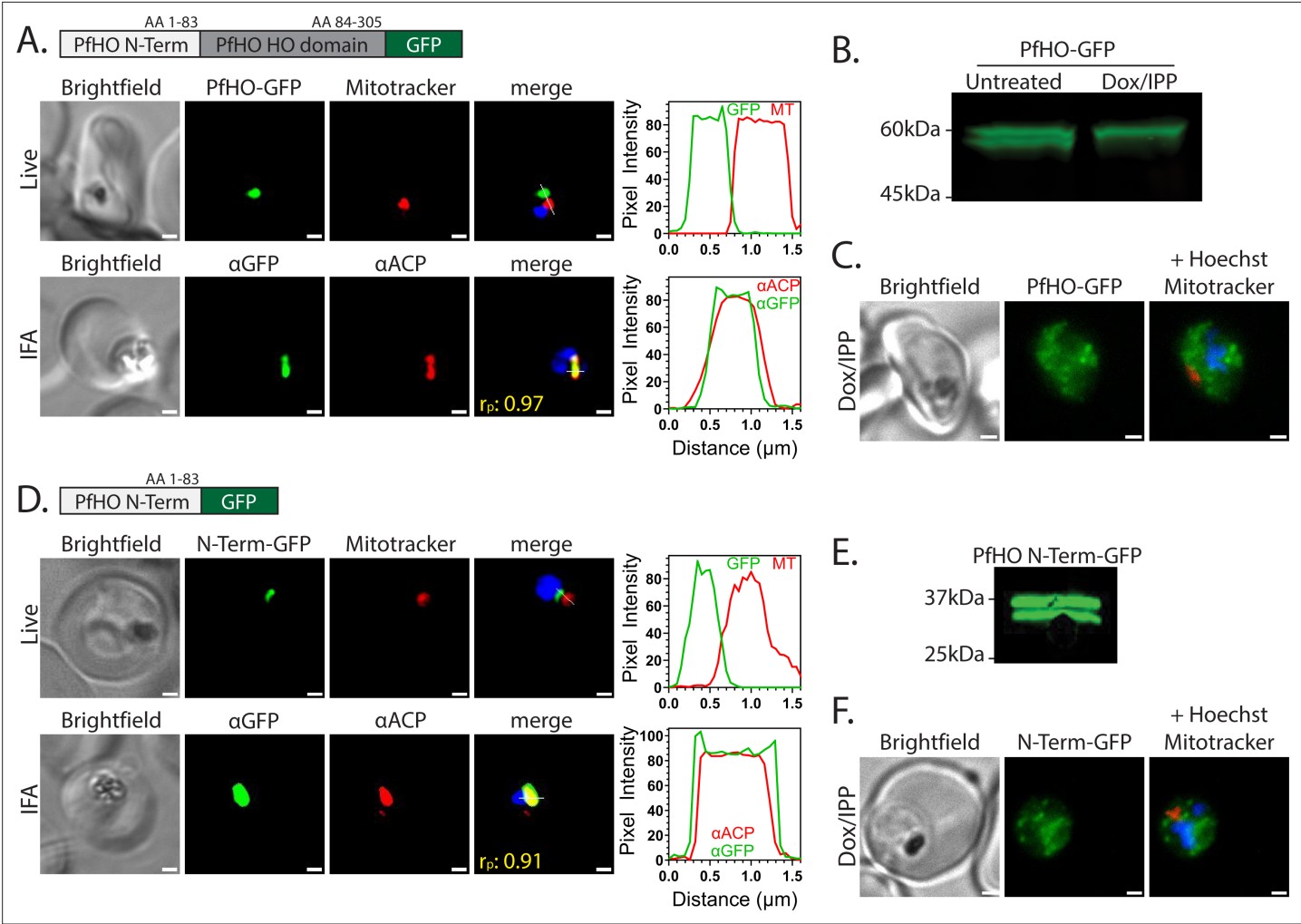

**Figure 2.** PfHO localization and processing. (**A**) Widefield fluorescence microscopy of Dd2 parasites episomally expressing PfHO-GFP. For live imaging, parasites were stained with 25 nM Mitotracker Red and 10 nM Hoechst. For IFA, parasites were fixed and stained with anti-GFP and anti-apicoplast ACP antibodies, as well as DAPI. For all images, the white scale bars indicate 1 μm. The average Pearson correlation coefficient ($r_p$) of red and green channels based on ≥10 images. Pixel intensity plots as a function of distance along the white line in merged images are displayed graphically beside each parasite. (**B**) Western blot of untreated or Dox/IPP-treated parasites episomally expressing PfHO-GFP and stained with anti-GFP antibody. (**C**) Live microscopy of PfHO-GFP parasites cultured 7 days in 1 μM doxycycline (Dox) and 200 μM IPP and stained with 25 nM Mitotracker Red and 10 nM Hoechst. (**D**) Widefield fluorescence microscopy of Dd2 parasites episomally expressing PfHO N-term-GFP and stained as in panel A. (**E**) Western blot of parasites episomally expressing PfHO N-term-GFP and stained with anti-GFP antibody. (**F**) Live microscopy of PfHO N-term-GFP parasites cultured 7 d in 1 μM Dox and 200 μM IPP, and stained as in panel C. For each parasite line and condition, ≥20 parasites were analyzed by live imaging and ≥10 parasites were analyzed by IFA.

The online version of this article includes the following source data and figure supplement(s) for figure 2:

**Source data 1.** Uncropped and labeled western blot for *Figure 2B*.

**Source data 2.** Original unlabeled file for western blot image in *Figure 2B*.

**Source data 3.** Uncropped and labeled western blot for *Figure 2E*.

**Source data 4.** Original unlabeled file for western blot image in *Figure 2E*.

**Figure supplement 1.** Additional widefield fluorescence microscopy of live Dd2 parasites episomally expressing PfHO-GFP or PfHO N-term-GFP.

**Figure supplement 2.** Additional widefield immunofluorescence microscopy of fixed Dd2 parasites episomally expressing PfHO-GFP and PfHO N-Term-GFP and stained with anti-GFP and anti-apicoplast acyl carrier protein (ACP) antibodies, and DAPI.

To directly test whether the N-terminal leader sequence of PfHO is sufficient for apicoplast targeting, we episomally expressed the PfHO N-terminus (residues 1–83) fused to a C-terminal GFP tag in Dd2 parasites and observed a nearly identical pattern of GFP signal in live and fixed parasites compared to full-length PfHO (*Figure 2D*, *Figure 2—figure supplement 1* and *Figure 2—figure supplement 2*), as well as both precursor and processed bands by western blot analysis (*Figure 2E*). In Dox/IPP-treated parasites, PfHO N-term-GFP signal appeared as dispersed fluorescent foci (*Figure 2F* and *Figure 2—figure supplement 1*). Based on these observations, we conclude that PfHO is targeted by its N-terminal leader sequence for import into the apicoplast where it undergoes proteolytic processing. This processing and previously reported protein associations for PfHO (*Mallari et al., 2014*) suggest targeting to the apicoplast matrix. Studies described below for endogenously tagged PfHO further support this conclusion.

## PfHO is essential for parasite viability and apicoplast biogenesis

To directly test PfHO essentiality in blood-stage parasites, we edited the PfHO gene to enable conditional knockdown. We first used restriction endonuclease-mediated integration (*Black et al., 1995*) or CRISPR/Cas9 to tag PfHO with either a C-terminal GFP-tag and the DHFR-destabilization domain (*Muralidharan et al., 2011*; *Armstrong and Goldberg, 2007*) in 3D7 (PM1 KO; *Liu et al., 2005*) parasites or a C-terminal dual hemagglutinin (HA$_2$) tag and the glmS ribozyme (*Prommana et al., 2013*; *Winkler et al., 2004*) in Dd2 parasites, respectively (*Figure 3—figure supplement 1*). Although neither system provided substantial downregulation of PfHO expression, we used the GFP-tagged parasites to confirm apicoplast targeting and processing of endogenous PfHO. Western blot analysis of parasites expressing PfHO-GFP-DHFR$_{DD}$ revealed two bands by anti-GFP staining and only a single precursor protein band upon apicoplast disruption in Dox/IPP conditions (*Figure 3A*). Additionally, co-localization of anti-GFP and anti-ACP staining by IFA confirmed apicoplast targeting of endogenous PfHO-GFP-DHFR$_{DD}$ (*Figure 3—figure supplement 2*). Immunogold transmission electron microscopy (TEM) of fixed parasites stained with anti-GFP and anti-ACP antibodies co-localized both proteins within a single multi-membrane compartment, as expected for targeting to the apicoplast. We noted that PfHO appeared to preferentially associate with the innermost apicoplast membrane, while ACP signal was distributed throughout the apicoplast matrix (*Figure 3B* and *Figure 3—figure supplement 3*).

We next used CRISPR/Cas9 to tag the PfHO gene in Dd2 parasites to encode the aptamer/TetR-DOZI system, which places protein expression under control of the non-toxic small molecule, anhydrotetracycline (aTC; *Goldfless et al., 2014*). The PfHO gene was edited to include both a single aptamer at the 5′ end and a 10 x aptamer cassette at the 3′ end (*Nasamu et al., 2021*; *Ganesan et al., 2016*) but without introducing an epitope tag in the encoded protein sequence. Correct integration into the PfHO locus was validated by genomic PCR and Southern blot (*Figure 3—figure supplement 4*). Because the protein was untagged, we made a custom rabbit polyclonal antibody that was raised against the HO-like domain of PfHO and selectively recognized PfHO expressed in parasites and *E. coli* (*Figure 3—figure supplement 5*). Using the aptamer-tagged parasites and this custom antibody, we performed western blot analysis to confirm detection of endogenous PfHO in +aTC conditions, including observation of precursor and processed bands. Critically, we observed that growth in -aTC conditions reduced PfHO levels by ≥80% across biological replicate samples (*Figure 3C* and *Figure 3—figure supplement 6*), indicating substantial downregulation of PfHO protein expression. PfHO mRNA levels were also selectively decreased by ~75% upon aTC washout, consistent with prior reports that TetR-DOZI association with transcripts targets mRNA to P-bodies for degradation (*Figure 3—figure supplement 6*; *Maruthi et al., 2020*; *Okada et al., 2022*; *García-Guerrero et al., 2024*). Because of the consistency and stringency of knockdown achieved by the PfHO-aptamer/TetR-DOZI system, all subsequent knockdown experiments were performed in this line.

By synchronous growth assay, we found that PfHO knockdown in -aTC conditions resulted in a severe growth defect and widespread parasite death in the third intraerythrocytic growth cycle (*Figure 3C* and *Figure 3—figure supplement 7*). Parasite growth was fully rescued by culture supplementation with the key apicoplast product, isopentenyl pyrophosphate (IPP) (*Yeh and DeRisi, 2011*), with IPP washout after 4 d in -aTC/+IPP conditions resulting in rapid parasite death (*Figure 3C*). These observations directly support the conclusion that PfHO is essential for blood-stage parasite viability and has a critical function within the apicoplast. To test if PfHO knockdown impacted apicoplast

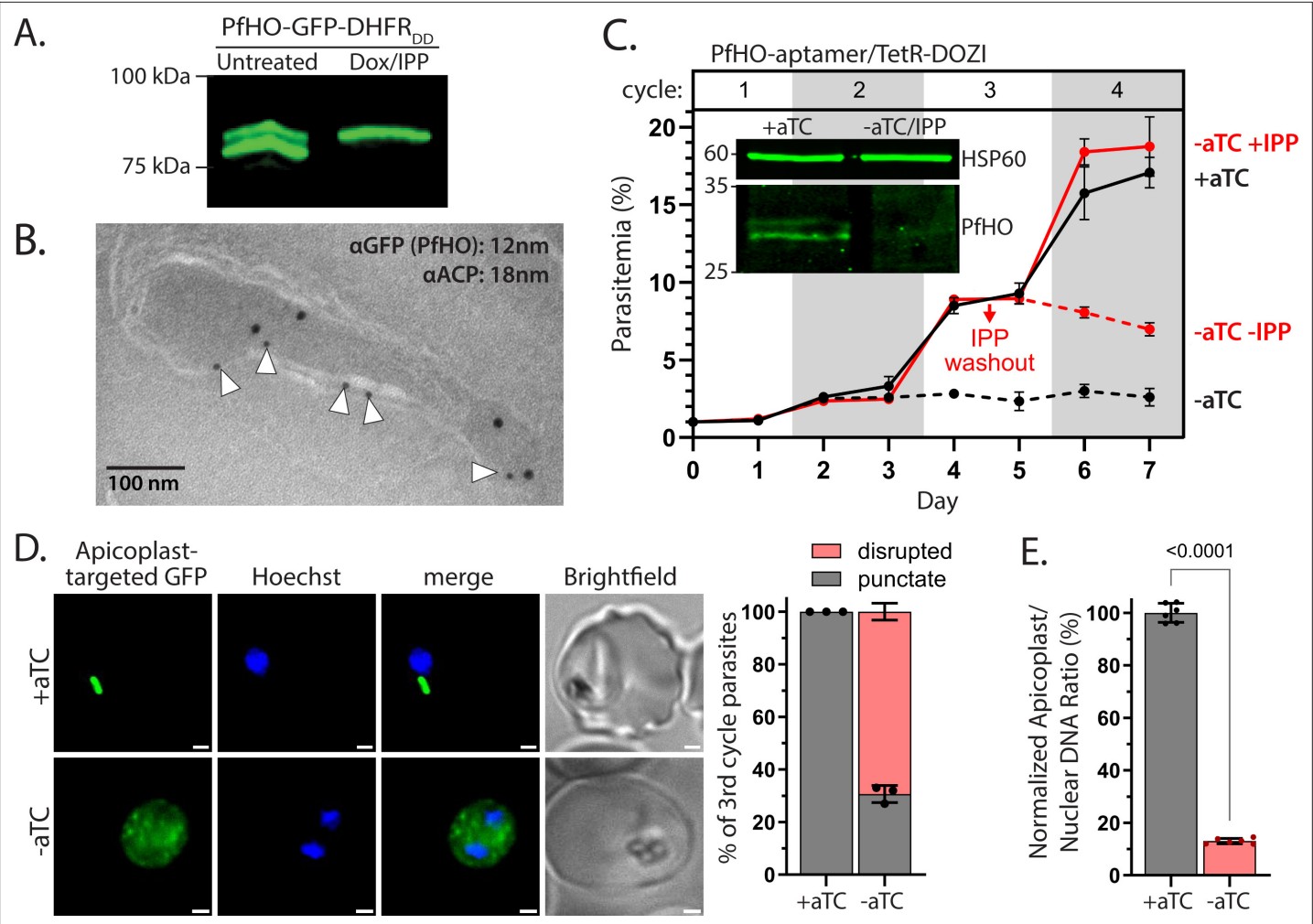

**Figure 3.** PfHO is essential for parasite viability and apicoplast maintenance. (**A**) Western blot of untreated or Dox/IPP-treated parasites with endogenously tagged PfHO-GFP-DHFR$_{DD}$. (**B**) Immunogold TEM of a fixed 3D7 parasite endogenously expressing PfHO-GFP-DHFR$_{DD}$ and stained with anti-GFP (12 nm, white arrows) and anti-apicoplast ACP (18 nm) antibodies. (**C**) Synchronized growth assay of Dd2 parasites tagged at the PfHO locus with the aptamer/TetR-DOZI system and grown ±1 μM aTC and ±200 μM IPP. Data points are the average ± SD of biological triplicates. Inset: western blot analysis of PfHO expression for 100 μg total lysates from parasites grown 3 d ±aTC, analyzed in duplicate samples run on the same gel, and stained with either custom anti-PfHO antibody or anti-heat shock protein 60 (HSP60) as loading control. Densitometry of western blot bands indicated >80% reduction in PfHO expression. (**D**) Live microscopy of PfHO-aptamer/TetR-DOZI parasites episomally expressing apicoplast-localized GFP (PfHO N-Term-GFP) grown 5 d ±aTC with 200 μM IPP. White scale bars in bottom right corners are 1 μm. Right: Population analysis of apicoplast morphology scored for punctate versus dispersed GFP signal in 110 total parasites from biological triplicate experiments. Statistical significance was calculated by Student's *t*-test. (**E**) Quantitative PCR analysis of the apicoplast: nuclear genome ratio for PfHO-aptamer/TetR-DOZI parasites cultured 5 d ±aTC with 200 μM IPP, based on amplification of apicoplast (SufB: Pf3D7_API04700, ClpM: Pf3D7_API03600, TufA: Pf3D7_API02900) relative to nuclear (STL: Pf3D7_0717700, I5P: Pf3D7_0802500, ADSL: Pf3D7_0206700) genes. Indicated qPCR ratios were normalized to +aTC and are the average ± SD of biological triplicates. Significance of ±aTC difference was analyzed by Student's *t*-test.

The online version of this article includes the following source data and figure supplement(s) for figure 3:

**Source data 1.** Uncropped and labeled western blot image of *Figure 3A*.

**Source data 2.** Original and unlabeled western blot image of *Figure 3A*.

**Source data 3.** Uncropped and labeled western blot images of *Figure 3C*.

**Source data 4.** Original and unlabeled western blot images of *Figure 3C*.

**Figure supplement 1.** Schemes for modification of the PfHO genomic locus to integrate C-terminal GFP-DHFR$_{DD}$ or HA$_2$-glmS tags.

**Figure supplement 1—source data 1.** Uncropped and labeled Southern blot image of *Figure 3—figure supplement 1B*.

**Figure supplement 1—source data 2.** Original and unlabeled Southern blot image of *Figure 3—figure supplement 1B*.

**Figure supplement 1—source data 3.** Uncropped and labeled PCR gel image of *Figure 3—figure supplement 1D*.

*Figure 3 continued on next page*

*Figure 3 continued*

**Figure supplement 1—source data 4.** Original and unlabeled PCR gel image of *Figure 3—figure supplement 1D*.

**Figure supplement 2.** Widefield immunofluorescence microscopy of fixed 3D7 parasites endogenously expressing PfHO-GFP-DHFR_{DD} and stained with anti-GFP and anti-apicoplast acyl carrier protein (ACP) antibodies, and DAPI.

**Figure supplement 3.** Additional immunogold transmission electron microscopy images of apicoplasts from fixed 3D7 parasite endogenously expressing PfHO-GFP-DHFR_{DD} and stained with anti-GFP (12 nM, green arrows) and anti-apicoplast ACP (18 nM) antibodies.

**Figure supplement 4.** Scheme for modification of the PfHO genomic locus to integrate the aptamer/TetR-DOZI system.

**Figure supplement 4—source data 1.** Uncropped and labeled PCR gel image of *Figure 3—figure supplement 4B*.

**Figure supplement 4—source data 2.** Original and unlabeled PCR gel image of *Figure 3—figure supplement 4B*.

**Figure supplement 4—source data 3.** Uncropped and labeled Southern blot image of *Figure 3—figure supplement 4C*.

**Figure supplement 4—source data 4.** Original and unlabeled Southern blot image of *Figure 3—figure supplement 4C*.

**Figure supplement 5.** Validation of custom PfHO antibody specificity.

**Figure supplement 5—source data 1.** Uncropped and labeled western blot images of *Figure 3—figure supplement 5A*.

**Figure supplement 5—source data 2.** Original and unlabeled western blot images of *Figure 3—figure supplement 5A*.

**Figure supplement 5—source data 3.** Original and unlabeled western blot images of *Figure 3—figure supplement 5B*.

**Figure supplement 6.** Quantitative PCR and additional western blot analysis of PfHO expression ±aTC with 200 µM IPP.

**Figure supplement 6—source data 1.** Uncropped and labeled western blot image of *Figure 3—figure supplement 6B*.

**Figure supplement 6—source data 2.** Original and unlabeled western blot image of *Figure 3—figure supplement 6B*.

**Figure supplement 7.** Giemsa-stained smears of PfHO-aptamer/TetR-DOZI parasites grown in ± aTC.

**Figure supplement 8.** Additional live-parasite fluorescence microscopy images of apicoplast morphology after PfHO knockdown.

biogenesis, we transfected PfHO-aptamer/TetR-DOZI parasites with the PfHO N-term-GFP episome to label the apicoplast. By widefield fluorescence microscopy, we observed that live parasites in +aTC conditions had focal GFP expression. In contrast, parasites cultured 5 d in -aTC/+IPP conditions displayed dispersed fluorescent foci in most parasites (*Figure 3D* and *Figure 3—figure supplement 8*). Using qPCR, we determined that parasites cultured 5 d in -aTC/+IPP conditions showed a dramatic reduction in apicoplast genomic DNA compared to parasites grown in +aTC conditions (*Figure 3E*). We conclude that PfHO function is essential for apicoplast biogenesis such that its knockdown (+IPP) results in parasite progeny that lack the intact organelle.

## Electropositive transit peptide of PfHO is largely retained after apicoplast import and required for essential function

Apicoplast-targeted proteins containing bipartite N-terminal leader sequences typically undergo proteolytic cleavage that fully or mostly removes the targeting peptide upon import into the organelle (*Waller et al., 2000*; *van Dooren et al., 2002*). Western blot analyses confirmed that PfHO is N-terminally processed (*Figure 2B* and *Figure 3A*), but we noted that the size of the mature protein was several kDa larger than the estimated size of PfHO^{84-305} which was previously studied as the mature HO-like domain (*Sigala et al., 2012*). Using the endogenously tagged PfHO-HA_2 (glmS) line, we observed that the mature protein migrated by SDS-PAGE/western blot with an apparent molecular mass of ~34 kDa while the HO-like domain (PfHO^{84-305}-HA_2) recombinantly expressed in *E. coli* migrated at ~31 kDa (*Figure 4A* and *Figure 3—figure supplement 4*). This observation strongly suggested that only a portion of the targeting sequence was removed upon apicoplast import and that additional N-terminal sequence beyond the HO-like domain was present in mature PfHO. Based on this approximate size difference, we estimated that ~30–40 residues of the apicoplast-targeting sequence upstream of residue 84 were likely retained in mature PfHO.

To specify the N-terminus of mature PfHO, we immunoprecipitated endogenous PfHO from parasites using the PfHO-specific antibody, fractionated the eluted sample by SDS-PAGE, transferred to PVDF membrane, and performed N-terminal protein sequencing of the Coomassie-stained band corresponding to mature PfHO. This analysis suggested an N-terminal sequence of GPLGYLNR, which corresponds to a single sequence starting at residue 33 within the electropositive transit peptide of PfHO (*Figure 1A*). Mass spectrometry analysis of PfHO protein purified from parasites and subjected to tryptic digest provided broad peptide coverage of PfHO sequence and identified GPLGYLNR as

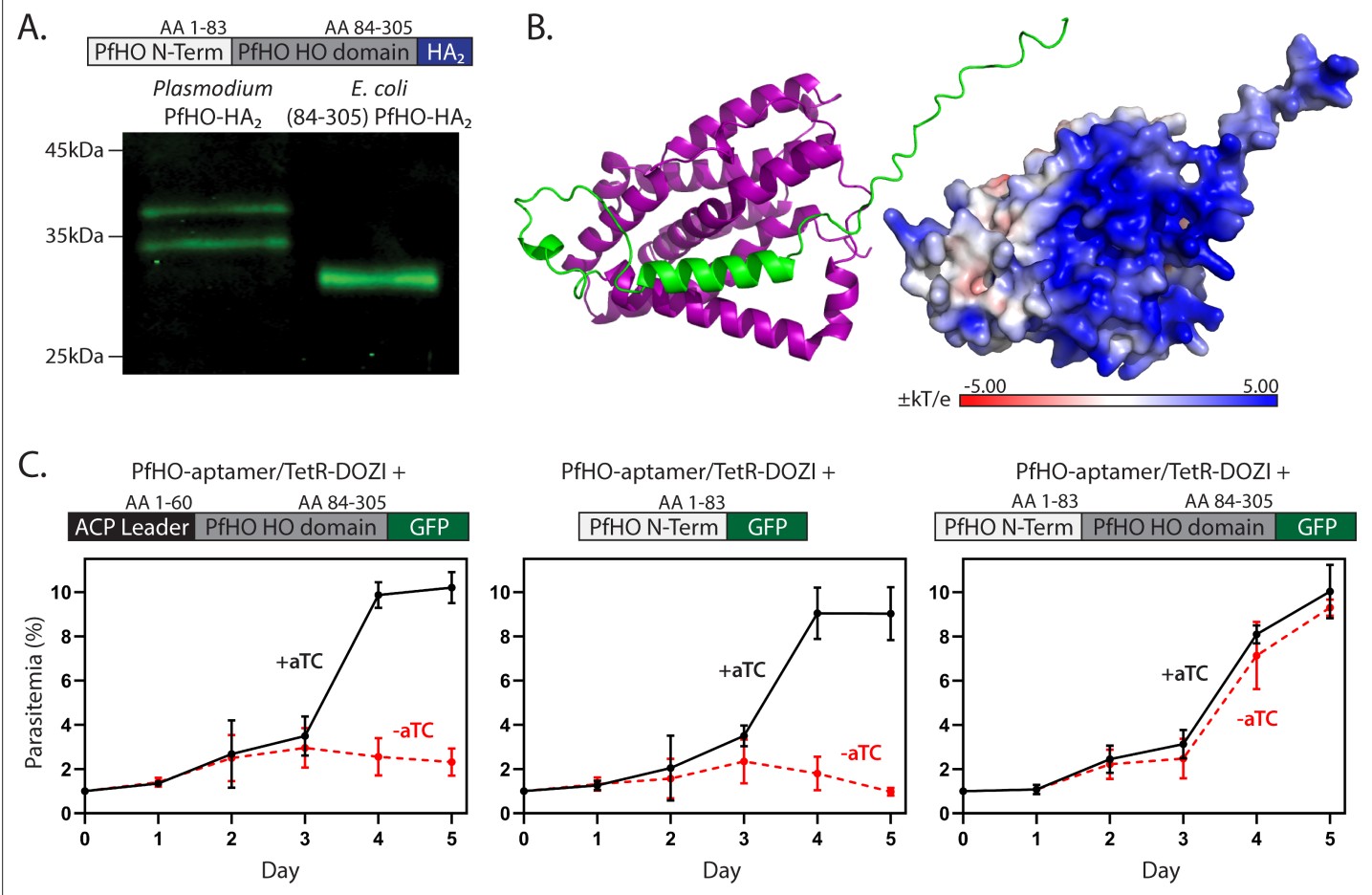

**Figure 4.** Processing and essentiality of the PfHO N-terminus. (**A**) Western blot of lysates from Dd2 parasites endogenously expressing PfHO-HA₂ and from *E. coli* recombinantly expressing PfHO$^{84-305}$-HA₂. (**B**) AlphaFold structure and electrostatic surface charge predicted for PfHO$^{33-305}$ corresponding to mature PfHO after apicoplast import. (**C**) Synchronized growth assays of PfHO knockdown Dd2 parasites complemented by episomal expression of the indicated PfHO constructs in ±1 μM aTC. Growth assay data points are the average ± SD of biological triplicates.

The online version of this article includes the following source data and figure supplement(s) for figure 4:

**Source data 1.** Uncropped and labeled western blot image of *Figure 4A*.

**Source data 2.** Original and unlabeled western blot image of *Figure 4A*.

**Figure supplement 1.** Peptide coverage of PfHO sequence detected by mass spectrometry.

**Figure supplement 1—source data 1.**

**Figure supplement 2.** qPCR analysis of PfHO knockdown in PfHO-aptamer/TetR-DOZI parasites complemented with indicated episomes.

**Figure supplement 3.** Western blot of PfHO-aptamer/TetR-DOZI parasites complemented with episomes expressing PfHO-GFP, ACP$_L$-HO-GFP, or PfHO N-Term-GFP.

**Figure supplement 3—source data 1.** Uncropped and labeled western blot image of *Figure 4—figure supplement 3*.

**Figure supplement 3—source data 2.** Original and unlabeled western blot image of *Figure 4—figure supplement 3*.

**Figure supplement 4.** Sequence alignment of alveolate HO-like proteins in *Plasmodium falciparum*, *Theileria orientalis*, *Babesia microti*, *Chromera velia*, and *Vitrella brassicaformis*.

the most N-terminal peptide that was detected (*Figure 4—figure supplement 1*). The calculated molecular mass of 35.3 kDa for PfHO$^{33-305}$-HA₂ is similar to the observed SDS-PAGE migration for mature PfHO-HA₂ ~34 kDa (*Figure 4A*). Based on these observations, we conclude that PfHO is proteolytically processed upon apicoplast import to remove part but not all of the targeting peptide and result in an N-terminus at or near Gly₃₃ in mature PfHO.

Cleavage before Gly₃₃ leaves ~50 residues of the electropositive targeting peptide attached to the HO-like domain of mature PfHO. Intrinsic structural disorder is a fundamental property of

apicoplast-targeting peptides (*Gallagher et al., 2011*). Consistent with overall structural heterogeneity in these ~50 residues, we were unable to crystallize recombinant PfHO[33-305] that matched the mature protein, despite the presence of the structured HO-like domain. Nevertheless, we note that AlphaFold predicts that residues 57–72 of the N-terminus form an α-helix that folds across the HO-like domain of PfHO between the proximal and distal helices (*Figure 4B*). Additionally, the abundance of Arg and Lys residues within the retained N-terminal sequence (*Figure 1A*) grants a strong electropositive character to the surface of mature PfHO (*Figure 4B*).

To test if this retained N-terminal sequence contributes to essential PfHO function beyond a role in apicoplast targeting, we performed complementation studies using PfHO knockdown parasites. We transfected the PfHO-aptamer/TetR-DOZI parasites with episomes encoding the PfHO N-terminus fused to GFP (PfHO[1-83]-GFP), the HO-like domain of PfHO fused to the apicoplast ACP leader sequence (1-60; *Waller et al., 2000*) on its N-terminus and GFP on its C-terminus (ACP[L]-PfHO[84-305]-GFP), or full-length PfHO-GFP. We first confirmed knockdown of endogenous PfHO under -aTC conditions and proper expression and processing of the episomally expressed proteins in these parasite lines (*Figure 2*, *Figure 4—figure supplement 2*, and *Figure 4—figure supplement 3*). Although all three proteins were correctly targeted to the apicoplast and proteolytically processed, only expression of full-length PfHO with cognate leader sequence rescued parasite growth from knockdown of endogenous PfHO (*Figure 4C*). We conclude that the retained N-terminal sequence of mature PfHO contributes to essential function beyond its role in apicoplast targeting.

## PfHO associates with the apicoplast genome and mediates apicoplast gene expression

HO enzymes associate with a range of protein-interaction partners that depend on the organism and functional context. Known HO interactors include ferredoxin in plants and bacteria (*Tohda et al., 2021*; *Sugishima et al., 2004*), cytochrome P450 reductase in mammals (*Wilks, 2002*; *Wang and de Montellano, 2003*), and direct or indirect interactions with transcription factors that impact nuclear gene expression in mammals (*Scaffa et al., 2022*; *Dennery, 2014*; *Wu et al., 2021*). To identify protein-interaction partners of PfHO in parasites that might give insight into its essential role in apicoplast biogenesis, we used anti-HA immunoprecipitation (IP) to isolate endogenous PfHO-HA[2] from parasites. Co-purifying proteins were identified by tryptic digest and tandem mass spectrometry (MS), then compared to protein interactors identified in negative-control samples containing HA-tagged mitochondrial proteins mACP (*Falekun et al., 2021*) or cyt *c* (*Espino-Sanchez et al., 2023*) to filter out non-specific interactions. In two independent experiments, 509 proteins co-purified with PfHO but were not detected in pulldowns of either mitochondrial control (*Figure 5—figure supplement 1*). These PfHO-specific interactors included a range of cellular proteins, including proteins targeted to the apicoplast.

Because our microscopy and biochemical studies indicated exclusive PfHO localization to the apicoplast, we focused our analysis on co-purifying proteins that were known to localize to this organelle from prior IP/MS studies (*Boucher et al., 2018*; *Mallari et al., 2014*; *Figure 5—figure supplement 2*). Of the 65 apicoplast-targeted proteins, 37 had annotated functions, and the majority were associated with nucleic acid metabolism (e.g. GyrA/B, PREX, RAP, PKII) or protein translation (e.g. EF-G/Tu/Ts, RPS1, RPL15) pathways (*Figure 5A* and *Figure 5—figure supplement 3*). The most highly enriched PfHO-specific interactor in both IP/MS experiments was an unannotated protein (Pf3D7_1025300) that contains a putative aspartyl protease domain and shows distant structural similarity to DNA damage-inducible protein (Ddi-1, P40087) – a ubiquitin-dependent protease associated with transcription factor processing (*Trempe et al., 2016*; *Koizumi et al., 2016*; *Lehrbach and Ruvkun, 2016*; *Figure 5—figure supplement 2*). These putative interactors are consistent with reports of non-canonical HO roles in gene expression (*Biswas et al., 2014*; *Hsu et al., 2017*; *Krzeptowski et al., 2021*), retention by mature PfHO of an electropositive N-terminus favorable for interacting with nucleic acids (*Figure 4B*), and apparent PfHO localization to the membrane periphery of the apicoplast lumen (*Figure 3B*) where the apicoplast genome, DNA replication factors, and ribosomes associate (*Köhler et al., 1997*; *Martins-Duarte et al., 2021*; *Lemgruber et al., 2013*). We thus considered it most likely that PfHO had an essential function in either apicoplast genome replication or DNA-dependent gene expression.

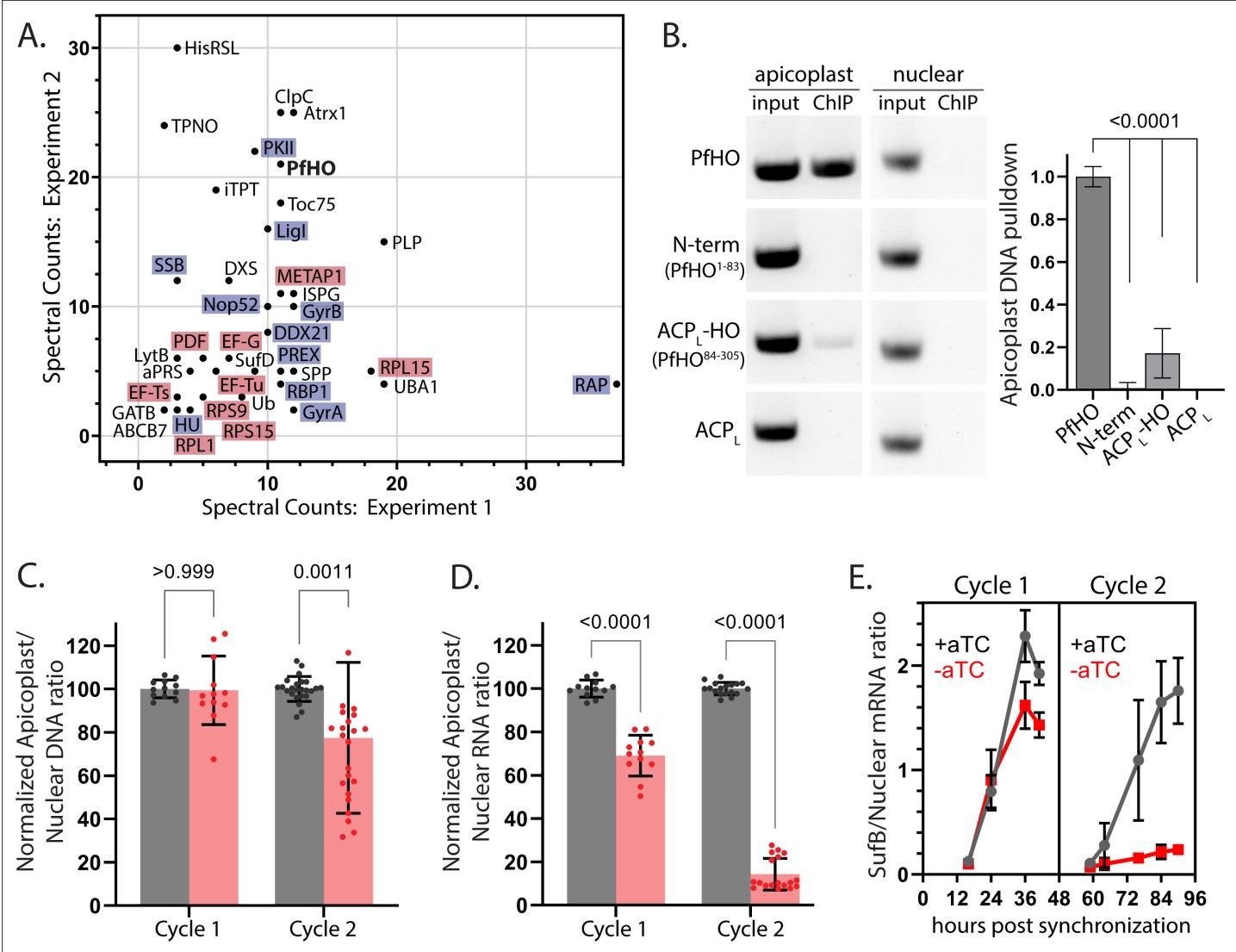

**Figure 5.** PfHO interactions with proteins and DNA and impacts of its knockdown on apicoplast DNA and RNA levels. (**A**) Spectral counts of functionally annotated, apicoplast-targeted proteins detected in two independent anti-HA IP/MS experiments on endogenously tagged PfHO-HA$_2$. The names of proteins in DNA/RNA metabolism are highlighted blue and proteins in translation are highlighted red. A list of all detected proteins can be found in *Figure 5—source data 1*. (**B**) Representative image showing PCR amplification of nuclear-encoded (apicoplast ACP: Pf3D7_0208500) and apicoplast-encoded (SufB: Pf3D7_API04700) genes from DNA co-purified with full-length PfHO-GFP, PfHO$^{1-83}$-GFP, ACP$_L$-PfHO$^{84-305}$-GFP, or ACP$_L$-GFP by αGFP ChIP. 'Input' is total parasite DNA collected after parasite lysis and sonication, and "ChIP" is DNA eluted after αGFP IP. Densitometry quantification of three biological replicates is plotted on right. Statistical significance of differences between PfHO and each other construct was calculated by Student's *t*-tests. (**C**) Quantitative PCR analysis of DNA isolated from tightly synchronized PfHO-aptamer/TetR-DOZI parasites grown ±1 μM aTC with 200 μM IPP and harvested at 36 and 84 hr in biological triplicates, with normalization of Ct values averaged from three apicoplast genes (SufB, TufA, ClpM) to Ct values averaged from three nuclear (STL, I5P, ADSL) genes. Grey bars represent +aTC and red bars represent –aTC, and observed ratios are displayed as percentages. (**D**) Quantitative RT-PCR on RNA isolated from the same parasites as in panel C to determine the normalized ratio of apicoplast transcripts (SufB, TufA, ClpM) relative to nuclear (STL, I5P, ADSL) transcripts. Significance of ±aTC differences for C and D were analyzed by Student's *t*-test. (**E**) Representative time-course showing Ct values of SufB normalized to three nuclear (STL, I5P, ADSL) genes at indicated time in PfHO-aptamer/TetR-DOZI parasites grown ±1 μM aTC with 200 μM IPP. Data points are the average ± SD of biological triplicates.

The online version of this article includes the following source data and figure supplement(s) for figure 5:

**Source data 1.** Table of proteins identified in PfHO IP/MS experiments.

**Source data 2.** Uncropped and labeled PCR gel images of *Figure 5B*, *Figure 5—figure supplement 5*.

**Source data 3.** Original and unlabeled PCR gel images of *Figure 5B*, *Figure 5—figure supplement 5*.

**Figure supplement 1.** Spectral counts for proteins that co-purified with PfHO and not with either mitochondrial control in both IP/MS experiments.

*Figure 5 continued on next page*

*Figure 5 continued*

**Figure supplement 2.** List of apicoplast-localized proteins co-purified with PfHO in two IP/MS experiments ordered by average of spectral counts for each experiment.

**Figure supplement 2—source data 1.**

**Figure supplement 3.** Functional pathway predictions for the 65 apicoplast-localized proteins that co-purified with PfHO in two IP/MS experiments.

**Figure supplement 4.** Quantification of parasite DNA fragment size by Agilent Bioanalyzer DNA analysis after pulse-sonication shearing of parasite lysates.

**Figure supplement 5.** Steady-state PCR amplification of additional nuclear (mACP: Pf3D7_1208300) and apicoplast (ClpM: Pf3D7_API03600) genes from DNA co-purified with indicated PfHO constructs.

**Figure supplement 6.** Additional ChIP experiments in parasites with GFP-tagged PfHO constructs.

**Figure supplement 6—source data 1.** Uncropped and labeled PCR gel image of *Figure 5—figure supplement 6D*.

**Figure supplement 6—source data 2.** Original and unlabeled PCR gel image of *Figure 5—figure supplement 6D*.

**Figure supplement 7.** RT-qPCR data for additional apicoplast genes.

**Figure supplement 8.** Representative time-course showing ClpM: nuclear transcript levels at indicated times during first and second cycle of parasite growth ±aTC in the presence of IPP.

To test the capability of PfHO to associate with the apicoplast genome, we leveraged an anti-GFP chromatin IP (ChIP) assay (*Sullivan and Santos, 2020*; *Wang et al., 2022a*) in parasites episomally expressing the GFP-tagged PfHO constructs tested in *Figure 4C*, or ACP$_L$-GFP as a negative control. We attempted to PCR or qPCR amplify multiple nuclear- and apicoplast-encoded genes in purified ChIP and input samples that had been sonicated to shear DNA into fragments ≤ 2 kb in size prior to IP (*Figure 5—figure supplement 4*). Target nuclear and apicoplast genes were both successfully amplified in all input samples. However, only the anti-GFP pulldown from parasites expressing full-length PfHO-GFP showed robust amplification of an apicoplast- but not nuclear-encoded gene (*Figure 5B*, *Figure 5—figure supplement 5*, and *Figure 5—figure supplement 6*). Although the portion of the PfHO N-terminus retained in mature PfHO has substantial electropositive character (*Figure 1A* and *Figure 4B*) favorable for association with DNA, this sequence in the PfHO$^{1-83}$-GFP construct was not sufficient for stable pull-down of apicoplast DNA. A faint amplicon for apicoplast DNA was detected for ACP$_L$-PfHO$^{84-305}$-GFP, but this signal was >fourfold weaker than observed for full-length PfHO (*Figure 5B*, *Figure 5—figure supplement 5*, and *Figure 5—figure supplement 6*). Based on these observations, we conclude that PfHO associates with the apicoplast genome and that DNA-binding requires both the cognate N-terminus and HO-like domain.

HO proteins in other species are reported to bind nuclear DNA (*Scaffa et al., 2022*), but the sequence features and nature of those associations are unclear. We note that our ChIP-PCR experiments cannot distinguish whether PfHO pull-down with the apicoplast genome reflects direct association with DNA and/or indirect interactions mediated by other proteins. Nevertheless, the unique sequence features of the PfHO N-terminus and its requirement for DNA association may suggest a *Plasmodium*-specific mechanism of DNA interaction that differs from other organisms. Selective interaction of full-length PfHO with the apicoplast genome was independent of the target gene amplified by PCR (*Figure 5—figure supplement 5*) or qPCR and persisted in the absence of crosslinking (*Figure 5—figure supplement 6*). Our observation that 12 distinct genes spanning the apicoplast genome show similar amplification in sheared PfHO ChIP samples (*Figure 5—figure supplement 6*) suggests that PfHO broadly interacts with apicoplast DNA in a sequence-independent manner akin to DNA topology regulators, gyrases, ligases, and single-strand stabilizing proteins (*Raghu Ram et al., 2007*; *Prusty et al., 2010*; *Buguliskis et al., 2007*), which our IP/MS data suggest are key interactors of PfHO (see discussion below).

Our observation that full-length PfHO, containing the cognate N-terminus and HO-like domain, was concordantly required for both DNA binding (*Figure 5B*) and essential function (*Figure 4C*) suggested most simply that association with the apicoplast genome was critical to PfHO function. To test possible roles for PfHO in DNA replication and/or RNA expression, we synchronized parasites to a 5 hr window and determined the impact of PfHO knockdown on DNA and RNA abundance by qPCR and RT-qPCR, respectively, in the first and second cycles after aTC washout but before apicoplast loss and parasite death. Parasites were grown in the presence of 200 μM IPP to decouple cellular viability from apicoplast-specific defects. We observed that PfHO knockdown in -aTC conditions caused a

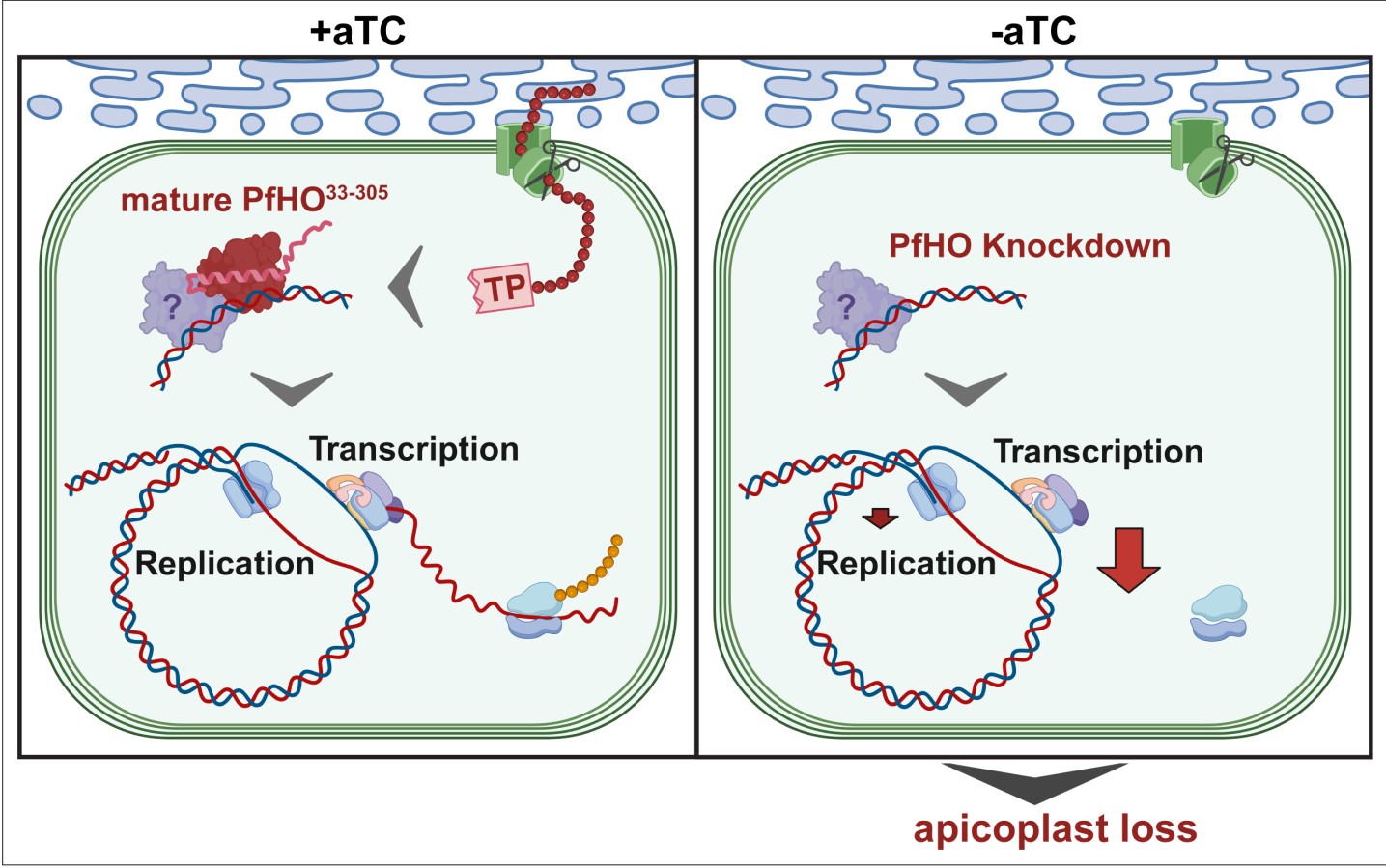

**Figure 6.** Model for essential PfHO function in apicoplast genome expression and organelle biogenesis. TP = transit peptide. Scissors represent proteolytic processing of the PfHO N-terminal TP upon apicoplast import.

modest ~20% decrease in apicoplast DNA levels in second-cycle parasites relative to +aTC conditions (*Figure 5C*). In contrast, apicoplast RNA levels were strongly reduced upon PfHO knockdown, with a 30% reduction observed in the first cycle and nearly 90% reduction in the second cycle after aTC washout (*Figure 5D*). This PfHO-dependent reduction in RNA abundance was observed for all tested protein-coding and non-coding apicoplast genes spanning all of the currently known or predicted polycistronic apicoplast transcripts (*Figure 5—figure supplement 7*; *Nisbet et al., 2016*; *Nisbet and McKenzie, 2016*; *Kobayashi et al., 2023*; *Denny et al., 1996*). In contrast to its impact on apicoplast RNA, PfHO knockdown had no measurable impact on RNA transcript abundance for nuclear or mitochondrial genes (*Figure 5—figure supplement 7*).

To dissect the time-course of this apicoplast-specific defect in RNA abundance, we collected samples from tightly synchronized parasites throughout the first and second growth cycles after aTC washout. In +aTC conditions, apicoplast RNA levels peaked around 36 hr for most genes, consistent with prior studies of apicoplast transcription (*Milton and Nelson, 2016*; *Painter et al., 2018*; *Llinás et al., 2006*). In -aTC conditions, there was a modest decrease in RNA transcript levels in the first cycle but complete failure to increase RNA abundance in the second cycle (*Figure 5E* and *Figure 5—figure supplement 8*). We conclude that PfHO function is essential for apicoplast gene expression and that PfHO knockdown results in a specific defect in RNA abundance that underpins higher order defects in apicoplast biogenesis that lead to parasite death (*Figure 6*).

## Discussion

Heme metabolism is a central cellular feature and critical therapeutic vulnerability of blood-stage malaria parasites, which have evolved unusual molecular adaptations to survive and grow within

heme-rich RBCs. Hemozoin is the dominant fate of labile heme released from large-scale hemoglobin digestion. Nevertheless, malaria parasites express a divergent and inactive heme oxygenase-like protein, whose cellular function underpinning its evolutionary retention has remained mysterious. We have elucidated an essential role for PfHO within the apicoplast organelle of *P. falciparum*, where it associates with the apicoplast genome and nucleic acid metabolism enzymes and is required for organelle gene expression and apicoplast biogenesis.

## Molecular function of PfHO

Our study unveils that *P. falciparum* parasites have repurposed the HO scaffold from its canonical role in heme degradation towards a divergent function required for expression of the apicoplast genome. This essential function appears to involve direct and/or indirect association with both the apicoplast genome and a variety of DNA/RNA-metabolism enzymes. Also, both the cognate N-terminal leader sequence, most of which remains attached in the mature protein, and the HO-like domain are required for function. We hypothesize that the electropositive N-terminus (*Figures 1A and 4B*) may mediate direct association with the apicoplast genome, while an electronegative face on the HO-like domain opposite the heme-binding region (*Figure 1—figure supplement 5*) may interact with other DNA-binding proteins (e.g. gyrases and helicases) which co-purified with PfHO in IP/MS studies (*Figure 5A*).

The specific molecular function of PfHO that impacts RNA transcript levels in the apicoplast remains to be defined, along with the broader prokaryotic-like biochemical processes that enable transcription in the apicoplast. The apicoplast genome is organized into two polycistronic operons that each consist of roughly half the genome, orient in opposite directions, and contain duplicated rRNA genes at their 5' ends (*Denny et al., 1996*). No promotor sequences have been identified in the apicoplast genome, but detection of transcripts up to 15 kb (*Nisbet et al., 2016*) and studies on mRNA processing sites (*Nisbet and McKenzie, 2016*) suggest that full polycistronic operons are transcribed and undergo 'punctuation processing' at tRNAs distributed throughout transcripts (*Dorrell et al., 2014*; *Ojala et al., 1981*). The primary subunits of a prokaryotic-like RNA polymerase complex have been identified in *Plasmodium*. Although associated sigma factors and other interacting proteins are proposed to exist, they have thus far remained difficult to identify (*Nisbet et al., 2016*; *Kobayashi et al., 2023*). We initially considered a model whereby PfHO might function as a sigma factor-like adaptor that directly binds to the multi-subunit prokaryotic RNA polymerase to mediate genomic association and transcription initiation. However, none of the core RNA polymerase subunits co-purified with PfHO in our IP/MS studies, suggesting that such a function may be unlikely.

Based on PfHO association with a variety of DNA-binding proteins and topology regulators, we consider it more likely that PfHO functions in association with other effector proteins to regulate and optimize DNA topology to enable transcriptional activity. The apicoplast-targeted gyrase A, gyrase B, DDX21 DEAD-box helicase, ligase I, the helicase domain of PREX, histone-like protein (HU) (*Ram et al., 2008*), and single-strand DNA binding protein (*Prusty et al., 2010*) all co-purify with PfHO in our IP/MS experiments (*Figure 5A*). Of these proteins, DNA gyrase A (*Augagneur et al., 2012*), gyrase B (*Pakosz et al., 2021*), PREX (*Lindner et al., 2011*), and HU *Sasaki et al., 2009* have been directly tested as essential apicoplast biogenesis factors in *Plasmodium* parasites. The relationship of most of these proteins with apicoplast transcription is unknown, but their activity in other organisms may shed light on a functional pathway in *Plasmodium*. Prokaryotic transcription is blocked by chemical or genetic disruption of DNA gyrase (*Oostra and Gruber, 1980*; *Gupta et al., 2006*; *Wahle et al., 1985*), a hetero-tetramer of gyrase A and B proteins (*Champoux, 2001*). DNA gyrase-induced negative supercoiling locally unwinds circular DNA molecules (*Wahle et al., 1985*) to facilitate transcriptional initiation by RNA polymerase complexes (*Oostra and Gruber, 1980*; *Ahmed et al., 2017*). DNA gyrase also relieves torsional stress formed by the procession of RNA polymerase complexes along DNA molecules and prevents stalling during transcription of long, polycistronic operons (*Dorman, 2019*; *Feagin and Drew, 1995*). Additionally, HU coordinates prokaryotic DNA structure and supercoiling in conjunction with DNA gyrase and has been reported to regulate the spatial distribution of RNA polymerase and transcription levels in *E. coli* (*Berger et al., 2010*; *Oberto et al., 2009*). Although these proteins have exclusively been associated with apicoplast DNA replication in *Plasmodium*, regulation of DNA supercoiling is also a major mechanism of prokaryotic transcriptional control (*Dorman, 2019*; *Dillon and Dorman, 2010*). This biological property may have been present in ancestral plastids and inherited by the prokaryotic-like *Plasmodium* apicoplast.

We also noted that RNA processing and translation-associated proteins co-purified with PfHO, including ribosomal RNA (rRNA) and transfer RNA (tRNA) maturation proteins, elongation factors, and ribosomal protein subunits (*Figure 5A*). RNA metabolism in the *Plasmodium* apicoplast is sparsely understood. No RNA-degrading enzymes have been identified, and the specific functions of RNA-binding proteins remain unknown. DEAD-box DNA/RNA helicases such as DDX21 have been implicated in the removal of aberrant R-loops (DNA/RNA hybrids) during RNA transcription (*Hao et al., 2024*; *Song et al., 2017*; *Saha et al., 2022*), but *Plasmodium* DDX21 has also been implicated with rRNA maturation in ribosome biogenesis (*Mallari et al., 2014*). Other RNA-binding proteins that co-purified with PfHO, Nop52 and RAP, also show low-level sequence homology to ribosome assembly factors (*Savino et al., 1999*; *Sanghai et al., 2023*). Prokaryotic ribosome assembly is a co-transcriptional process regulated by, and in close proximity to RNA transcriptional machinery (*Davis and Williamson, 2017*; *Paul et al., 2004*; *Jourdain et al., 2017*). Apicoplast ribosome assembly is poorly studied but appears to be similar to prokaryotic systems. Indeed, a prior IP/MS study reported that apicoplast ribosomes and ribosomal assembly complexes co-purified with RNA transcription complexes and DNA topology regulators (*Mallari et al., 2014*). PfHO pulldown with RNA-processing and protein-translation components may therefore reflect the underlying physical and temporal coupling of RNA transcription and protein translation in the apicoplast. It is also possible that PfHO contributes to other aspects of RNA metabolism that remain undefined.

Translation of apicoplast-encoded proteins is required for organelle biogenesis and inheritance (*Dahl et al., 2006*; *Dahl and Rosenthal, 2007*; *Dahl and Rosenthal, 2008*), including a likely essential role for the apicoplast-encoded chaperone ClpM in import of nuclear-encoded apicoplast-targeted proteins (*Florentin et al., 2017*; *Janouškovec et al., 2015*; *El Bakkouri et al., 2010*). Apicoplast-encoded SufB is also expected to be essential for synthesis of Fe-S clusters by the apicoplast SUF pathway that are required for IPP synthesis and organelle maintenance (*Okada and Sigala, 2023*; *Swift et al., 2023*; *Charan et al., 2014*). Therefore, we predict that apicoplast biogenesis defects resulting from PfHO knockdown are due to significantly diminished levels of apicoplast-encoded ribosomal and messenger RNAs required for apicoplast translation of ClpM and SufB.

## Evolution of PfHO and its divergent function

Although PfHO retains the conserved α-helical structure of HO enzymes, it has strikingly low sequence similarity, implying substantial evolutionary distance from well-studied HO orthologs in humans, plants, and bacteria (*Figure 1* and *Figure 1—figure supplement 2*). What then is the evolutionary origin of PfHO and its non-canonical function? The *Plasmodium* apicoplast is thought to derive from secondary endosymbiosis through ancestral engulfment of a plastid-containing red algae that had previously engulfed a photosynthetic cyanobacterium (*Köhler et al., 1997*; *Boucher and Yeh, 2019*; *Sanchez-Puerta and Delwiche, 2008*; *Keeling, 2013*; *McFadden et al., 1996*). Subsequent loss of photosynthesis accompanied the transition to intracellular parasitism by proto-apicomplexan ancestors (*Janouškovec et al., 2015*; *Woo et al., 2015*). HO enzymes are commonly found in photosynthetic cyanobacteria and eukaryotic chloroplasts, where they initiate biosynthesis of biliverdin and other bilin chromophores utilized in phytochrome proteins for light sensing and signaling (*Davis et al., 2001*; *Emborg et al., 2006*; *Aoki et al., 2011*). Retention of PfHO in the *Plasmodium* apicoplast likely reflects the original presence of HO enzymes in the cyanobacterial ancestors of this organelle.

The transition from free-living algae to apicomplexan parasitism involved significant genome reduction, including loss of plastid photosynthesis and phytochrome biosynthesis pathways (*Woo et al., 2015*). These functional reductions presumably removed the selective pressure to retain enzymatic HO activity. Indeed, the only identifiable HO homologs retained in Apicomplexa are found exclusively in hematozoan parasites such as *Plasmodium*, *Babesia*, and *Theileria*, and have lost the active-site His ligand like PfHO (*Figure 1—figure supplement 2*). Insights into the evolutionary origin of PfHO are provided by comparison to HO-like proteins in coral-symbiotic chromerid algae, the closest free-living and photosynthetic relatives to apicomplexan parasites (*Janouškovec et al., 2015*; *Woo et al., 2015*). *Vitrella brassicaformis* and *Chromera velia* both express multiple HO homologs thought to be remnants of prior endosymbioses (*Oborník and Lukeš, 2013*). These HO proteins segregate phylogenetically with either active metazoan, active plant, or inactive hematozoan HOs (*Figure 1—figure supplement 2*) featuring the retention or loss of the active-site His ligand, respectively. Similarly, *Arabidopsis thaliana* expresses four chloroplast-targeted HO homologs that include three active HO

enzymes and the inactive AtHO2 that lacks the conserved His ligand but contributes to photomorphogenesis (*Davis et al., 2001*). These observations support a model in which active HO homologs were lost along with photosynthetic machinery during the transition to apicomplexan parasitism with the exclusive retention of an inactive HO-like homolog in Hematozoa.

It remains unclear why hematozoan but no other apicomplexan parasites retained an inactive HO-like homolog. The common infection of heme-rich RBCs by hematozoans may suggest that remnant heme- or porphyrin-binding activity (*Sigala et al., 2012*) may play a role in functional regulation of these HO-like proteins within the apicoplast. Indeed, apicoplast transcription (*Painter et al., 2018*), heme released by hemoglobin digestion (*Goldberg et al., 1990*), and PfHO expression (*Figure 3—figure supplement 5*) all peak coincidentally around 30 hr post-infection. The affinity of PfHO for heme ($K_d$ ~8 μM; *Sigala et al., 2012*) is also notably similar to the ~2 μM concentration of labile heme estimated in the parasite cytoplasm (*Abshire et al., 2017*). Future studies can test the intriguing hypothesis that labile heme levels sensed by PfHO in the apicoplast tune functional interactions by PfHO that regulate apicoplast gene expression.

Although PfHO has diverged from canonical HO function, it retains many structural and biochemical features present in metazoan HOs which are reported to mediate transcription factor activity independent of heme degradation (*Mascaró et al., 2021*; *Lin et al., 2007*; *Jagadeesh et al., 2022*; *Vanella et al., 2016*; *Scaffa et al., 2022*). Thus, a role for PfHO in gene expression may be an ancient functional property of the HO scaffold that was further expanded and honed by parasites after heme-degrading activity was lost. In this regard, PfHO may be conceptually similar to other parasite proteins (e.g. mitochondrial acyl carrier protein *Falekun et al., 2021*) that have lost canonical function but whose retention of an essential role unveils a latent secondary activity that was previously un- or under-appreciated in the shadow of the dominant primary function in well-studied orthologs from other organisms.

## Implications for expanded functions of N-terminal pre-sequences beyond organelle targeting

Our discovery that PfHO requires a portion of its N-terminal transit peptide for essential function within the apicoplast expands the molecular paradigm for understanding the evolution and function of organelle-targeting leader sequences. PfHO homologs in *Vitrella* and *Chromera* also contain a predicted α-helix in their N-terminal sequences that differ from chloroplast-targeting HOs in plants (*Figure 4—figure supplement 4*). This unique α-helix predicted in alveolate HO-like proteins adds positive-charge to the HO-like protein that we propose mediates interaction with DNA. It is possible that the alveolate ancestors of *Plasmodium* expanded and functionally repurposed the targeting sequences of other plastid and/or mitochondrial proteins to provide unique organelle-specific functions that remain to be discovered. The recent report that *Toxoplasma gondii* parasites repurpose the cleaved leader sequence of mitochondrial cyt $c_1$ as a stable subunit of ATP synthase supports this view (*Maclean et al., 2021*; *Mühleip et al., 2021*). Identifying adaptations in *P. falciparum* that diverge from human host cells can reveal novel parasite vulnerabilities that underpin the development of new therapeutic strategies.

# Materials and methods

**Key resources table**

| Reagent type (species) or resource | Designation | Source or reference | Identifiers | Additional information |
|---|---|---|---|---|
| Cell line (*Plasmodium falciparum*) | Dd2 | PMID:1970614 | | |
| Cell line (*Plasmodium falciparum*) | Dd2 +PfHO(1-305)-GFP/TEOE | this paper | PfHO-GFP | randomly integrated with pHTH (PMID:16260745), Can be obtained from Sigala lab |
| Cell line (*Plasmodium falciparum*) | Dd2 +PfHO(1-83)-GFP/TEOE | this paper | N-term-GFP | randomly integrated with pHTH (PMID:16260745), Can be obtained from Sigala lab |

*Continued on next page*

*Continued*

| Reagent type (species) or resource | Designation | Source or reference | Identifiers | Additional information |
|---|---|---|---|---|
| Cell line (*Plasmodium falciparum*) | 3D7 - PfHO-DHFR$_{DD}$-GFP | this paper | | Can be obtained from Sigala lab |
| Cell line (*Plasmodium falciparum*) | Dd2 - PfHO-*GlmS*-HA$_2$ | this paper | | Can be obtained from Sigala lab |
| Cell line (*Plasmodium falciparum*) | Dd2 - PfHO-aptamer/TetR-DOZI | this paper | PfHO-Aptamer/TetR-DOZI | Can be obtained from Sigala lab |
| Cell line (*Plasmodium falciparum*) | Dd2 - PfHO-aptamer/TetR-DOZI+PfHO(1-83)-GFP/TEOE | this paper | PfHO-Apt +N-term-GFP | Can be obtained from Sigala lab |
| Cell line (*Plasmodium falciparum*) | Dd2 - PfHO-aptamer/TetR-DOZI+PfHO(84-305)-GFP/TEOE | this paper | PfHO-Apt +ACPL-HO-GFP | Can be obtained from Sigala lab |
| Cell line (*Plasmodium falciparum*) | Dd2 - PfHO-aptamer/TetR-DOZI+PfHO(1-305)-GFP/TEOE | this paper | PfHO-Apt +PfHO-GFP | Can be obtained from Sigala lab |
| Strain, strain background (*Escherichia coli*) | BL21(DE3) +PfHO(Δ2–83)-His$_6$/pET21d | PMID:22992734 | | |
| Strain, strain background (*Escherichia coli*) | BL21(DE3) + His$_6$-PfHO(Δ2–83)/pET28a | PMID:22992734 | | |
| Strain, strain background (*Escherichia coli*) | BL21(DE3) +PfHO(Δ2–83) HA$_2$/pET28a | this paper | | Can be obtained from Sigala lab |
| Chemical compound, drug | ampicillin | Sigma-Aldrich | Cat. No. A9518 | 50 µg/mL |
| Chemical compound, drug | isopropyl β-D-1-thiogalactopyranoside | Sigma-Aldrich | Cat. No. 16758 | 0.5 mM |
| Chemical compound, drug | Ni_NTA agarose column | Qiagen | Cat. No. 30210 | |
| Chemical compound, drug | RPMI-1640 | Thermo Fisher | Cat. No. 23400021 | |
| Chemical compound, drug | Albumax I Lipid-Rich BSA | Thermo Fisher | Cat. No. 11020039 | 2.5 g/L |
| Chemical compound, drug | anhyrdrotetracycline | Cayman Chemicals | Cat. No. 10009542 | 1 µM |
| Chemical compound, drug | doxycycline | Sigma-Aldrich | Cat. No. D3447 | 1 µM |
| Chemical compound, drug | isopentenyl pyrophosphate | Isoprenoids | Cat. No. IPP001 | 200 µM |
| Chemical compound, drug | D-sorbitol | Sigma-Aldrich | Cat. No. S7900 | 5% w/v |
| Commercial assay or kit | MACS LD Columns | Miltenyi Biotec | Cat. No. 130-042-901 | |
| Commercial assay or kit | NEBuilder HIFI DNA Assembly Mix | New England Biolabs | Cat. No. E2621 | |
| Chemical compound, drug | WR99210 | Jacobus Pharmaceutical Co | | 5 nM |
| Chemical compound, drug | Blasticidin S | Sigma-Aldrich | Cat. No. 15205 | 6 µM |
| Chemical compound, drug | DSM1 | Sigma-Aldrich | Cat. No. 53330401 | 2 µM |
| Chemical compound, drug | acridine orange | Invitrogen | Cat. No. A3568 | 1 µg/mL |
| Chemical compound, drug | Mitotracker Red CMXRos | Invitrogen | Cat. No. M7512 | 25 nM |

*Continued*

| Reagent type (species) or resource | Designation | Source or reference | Identifiers | Additional information |
|---|---|---|---|---|
| Chemical compound, drug | Hoechst 33342 | Invitrogen | Cat. No. H3570 | 5 µg/mL |
| Chemical compound, drug | Prolong Diamond Antifade Mountant with DAPI | Invitrogen | Cat. No. P36971 | |
| Chemical compound, drug | protease inhibitor tablets | Invitrogen | Cat. No. A32955 | |
| Chemical compound, drug | saponin from quillaja bark | Sigma-Aldrich | Cat. No. S7900 | |
| Commercial assay or kit | protein A dynabeads | Invitrogen | Cat. No. 1001D | |
| Chemical compound, drug | acridine orange | Invitrogen | Cat. No. A3568 | |
| Commercial assay or kit | QIAamp DNA Blood Mini kit | Qiagen | Cat. No. 51106 | |
| Commercial assay or kit | TRIzol reagent | Invitrogen | Cat. No. 15596026 | |
| Commercial assay or kit | Superscript IV VILO RT kit | Invitrogen | Cat. No. 11766050 | |
| Chemical compound, drug | Proteinase K | Invitrogen | Cat. No. 25530049 | |
| Commercial assay or kit | Qiaquick PCR purification kit | Qiagen | Cat. No. 28104 | |
| Chemical compound, drug | salmon sperm DNA | Invitrogen | Cat. No. AM9680 | |
| Commercial assay or kit | Prometheus ProSignal Femto ECL reagent | Genesee Scientific | Cat. No. 20–302 | |
| Antibody | anti-GFP (goat, polyclonal) | Abcam | Cat. No ab5450 | WB (1:1000), IFA (1:200) |
| Antibody | anti-GFP (mouse, monoclonal 3E6) | Invitrogen | Cat. No. A-11120 | IFA (1:200) |
| Antibody | anti-HA (rat, monoclonal 3F10) | Roche | Cat. No 11-867-423-01 | WB (1:1000), IFA (1:200) |
| Antibody | anti-ACP (rabbit, polyclonal) | PMID:19768685 | | IFA (1:200) |
| Antibody | anti-EF1α (rabbit, polyclonal) | PMID:11251817 | | WB (1:1000) |
| Antibody | anti-PfHO (rabbit, polyclonal) | This study | | WB (1:500), Can be obtained from Sigala lab |
| Antibody | anti-goat HRP (rabbit, polyclonal) | Santa Cruz Biotechnology | Cat. No. sc-2768 | WB (1:10,000) |
| Antibody | anti-rabbit HRP (goat, polyclonal) | Invitrogen | Cat. No. A-27036 | WB (1:10,000) |
| Antibody | anti-rabbit IRDye800CW (donkey, polyclonal) | LiCor | Cat. No. 926–32213 | WB (1:10,000) |
| Antibody | anti-rabbit IRDye680 (donkey, polyclonal) | LiCor | Cat. No. 926–68023 | WB (1:10,000) |
| Antibody | anti-goat IRDye800CW (donkey, polyclonal) | LiCor | Cat. No. 926–32214 | WB (1:10,000) |
| Antibody | anti-rat IRDye800CW (goat, polyclonal) | LiCor | Cat. No. 926–32219 | WB (1:10,000) |
| Antibody | anti-mouse AF488 (goat, polyclonal) | Invitrogen | Cat. No. A-11001 | IFA (1:1000) |
| Antibody | anti-rabbit AF647 (goat, polyclonal) | Invitrogen | Cat. No. A-21244 | IFA (1:1000) |
| Antibody | anti-goat AF488 (donkey, polyclonal) | Abcam | Cat. No. ab150129 | IFA (1:1000) |
| Software, algorithm | Prism 9 | GraphPad | RRID:SCR_002798 | |

## Sequence homology searches and phylogenetic analyses

We acquired *Plasmodium* and alveolate protein sequences from VEuPathDB.org (*Amos et al., 2022*) and all other protein sequences from https://www.UniProt.org; *Bateman et al., 2023* databases. The

protein sequence for PfHO (Pf3D7_1011900) was analyzed by NCBI Protein BLAST (*Altschul et al., 1990*) and HMMER (*Potter et al., 2018*) with the exclusion of *Plasmodium* or apicomplexan organisms to identify putative orthologs. Orthologous protein sequences and select reference HO proteins were aligned via Clustal Omega (*Sievers and Higgins, 2014*) and analyzed using Jalview (*Waterhouse et al., 2009*). The multi-sequence alignment was uploaded to the IQ-TREE webserver (http://iqtree.cibiv.univie.ac.at) with ultrafast bootstrap analysis. The resulting maximum likelihood phylogenetic tree from 1000 bootstrap alignments was analyzed and displayed using FigTree (http://tree.bio.ed.ac.uk/software/figtree/).

## Recombinant protein expression and purification for crystal structure determination

The gene encoding residues 84–305 of PfHO was cloned into a pET21d expression vector (Novagen) using NcoI and XhoI sites, in frame with a C-terminal His$_6$ tag (*Sigala et al., 2012*). *E. coli* BL21 (DE3) cells transformed with this vector were grown in LB medium supplemented with ampicillin (50 µg/mL; Sigma A9518) and protein expression was induced with 0.5 mM isopropyl β-D-1-thiogalactopyranoside (IPTG) (Sigma 16758) at an OD of 0.5, after which the cells were grown at 20 °C overnight. Cells were harvested by centrifugation (7000×$g$, 10 min), the pellet was resuspended in binding buffer (20 mM Tris-HCl, 500 mM NaCl, 20 mM imidazole, pH 8.5), and the cells were lysed by sonication. The cell lysate was cleared by centrifugation (40,000 × $g$, 30 min), and the supernatant was subjected to immobilized metal affinity chromatography using a 1 mL Ni-NTA agarose column (QIAGEN, 30210) equilibrated with binding buffer. The protein was eluted with a buffer containing 20 mM Tris-HCl, 500 mM NaCl, and 300 mM imidazole at pH 8.5. The eluted protein was further purified by size-exclusion chromatography employing a 26/60 Superdex75 column equilibrated with a buffer containing 20 mM HEPES and 300 mM NaCl at pH 7.5. To produce the protein containing SeMet, *E. coli* B834 (DE3) cells transformed with the same plasmid as above were grown in minimal media (2 g L$^{-1}$ NH$_4$Cl, 6 g L$^{-1}$ KH$_2$PO$_4$, 17 g L$^{-1}$ Na$_2$HPO$_4$·12H$_2$O, 1 g L$^{-1}$ NaCl, 1.6 mg L$^{-1}$ FeCl$_3$, 0.5 g L$^{-1}$ MgSO$_4$·7H$_2$O, 22 mg L$^{-1}$ CaCl$_2$·6H$_2$O, 4 g L$^{-1}$ Glucose, and 50 mg L$^{-1}$ L-SeMet). Purification was identical to that of the WT protein. Protein purity was confirmed by observation of a single band at the appropriate molecular mass by Coomassie-stained SDS-PAGE.

## Protein crystallization

Protein in a buffer containing 20 mM HEPES and 300 mM NaCl at pH 7.5 was subjected to crystallization trials using sitting-drop vapour diffusion using commercially available screening kits form Hampton Research in an Oryx8 system (Douglas Instruments). Protein (0.5 µL at 9.5 mg mL$^{-1}$) was mixed with an equal volume of reservoir solution, and crystallization plates were maintained at 20 °C for several weeks while being examined. The crystallization solutions producing the best crystals were optimized using hanging-drop geometry in 24-well plates by mixing manually 2 µL of protein solution (5.0 mg mL$^{-1}$) and an equal volume of reservoir solution. The best crystals appeared in a few days in a reservoir solution containing 0.4 M (NH$_4$)$_2$SO$_4$, 0.65 M Li$_2$SO$_4$, and 0.1 M sodium citrate tribasic dihydrate at pH 5.6 and a temperature of 20 °C. Single crystals were mounted in nylon Cryo-Loops (Hampton Research, HR4-932) coated with Paratone (Hampton Research, HR2-862) and directly transferred to liquid nitrogen for storage.

## Structural data collection and processing

Diffraction data from single crystals of WT and SeMet protein were collected in beamlines AR-NW12A and BL5A, respectively, at the Photon Factory (Tsukuba, Japan) under cryogenic conditions (100 K). Diffraction images were processed with the program MOSFLM and merged and scaled with the program SCALA or AIMLESS (*Evans, 2006*) of the CCP4 suite (*Winn et al., 2011*). The structure of the SeMet protein was solved by the method of single anomalous diffraction using the Autosol module included in the PHENIX suite (*Adams et al., 2010*). The structure of the WT protein was determined by the molecular replacement method using the coordinates of the SeMet protein from above with the program PHASER (*McCoy et al., 2007*). The models were refined with the programs REFMAC5 (*Murshudov et al., 1997*) and built manually with COOT (*Emsley et al., 2010*). Validation was carried out with PROCHECK (*Laskowski et al., 1993*). Data collection and structure refinement statistics are

given in *Figure 1—figure supplement 3*. The final structural coordinates and structure factors were deposited as RCSB Protein Data Bank entry 8ZLD.

## Structural visualization and analyses

A predicted structural model for PfHO (Pf3D7_1011900) was acquired from the AlphaFold Protein Structure Database (https://alphafold.ebi.ac.uk), and published HO structures were acquired from the RCSB Protein Data Bank (https://www.rcsb.org). The AlphaFold structural model, PfHO crystal structure, and HO structures were visualized and analyzed using The PyMOL Molecular Graphics System, Version 2.5, Schrödinger, LLC. Structural superpositions were performed with the PyMOL integrated command 'align' and assessed by the total number of atoms aligned and RMSD (Å). We uploaded PDB files of PfHO crystal structure and AlphaFold model to the DALI protein structural comparison server (http://ekhidna2.biocenter.helsinki.fi/dali) to identify proteins structurally related to PfHO. To determine surface charge of structures, we uploaded PDB files to APBS-PDB2PQR online software suite (https://server.poissonboltzmann.org/; *Jurrus et al., 2018*), and displayed the calculated Poisson-Boltzmann surface charge using the PyMOL APBS tool 2.1 plugin.

## Parasite culturing and transfection

*Plasmodium falciparum* Dd2 (*Wellems et al., 1990*) or 3D7 (*Liu et al., 2005*) parasites were cultured in human erythrocytes obtained from Barnes-Jewish Hospital (St. Louis, MO) or the University of Utah Hospital blood bank (Salt Lake City, UT) in RPMI-1640 medium (Thermo Fisher, 23400021) supplemented with 2.5 g/L Albumax I Lipid-Rich BSA (Thermo Fisher, 11020039), 15 mg/L hypoxanthine (Sigma, H9636), 110 mg/L sodium pyruvate (Sigma, P5280), 1.19 g/L HEPES (Sigma, H4034), 2.52 g/L sodium bicarbonate (Sigma, S5761), 2 g/L glucose (Sigma, G7021), and 10 mg/L gentamicin (Invitrogen, 15750060), as previously described (*Okada et al., 2022*). Parasites were maintained at 37 °C in 90% $N_2$/5% $CO_2$/5% $O_2$ or in 5% $CO_2$/95% air. For drug-induced apicoplast-disruption experiments, parasites were cultured for ~7 d in 1 µM doxycycline (Sigma, D9891) and 200 µM isopentenyl pyrophosphate (Isoprenoids, IPP001).

Transfections were performed in 1 x cytomix containing 50–100 µg DNA by electroporation of parasite-infected RBCs in 0.2 cm cuvetes using a Bio-Rad Gene Pulser Xcell system (0.31 kV, 925 µF). Transgenic parasites were selected based on plasmid resistance cassettes encoding human DHFR (*Fidock and Wellems, 1997*), yeast DHOD (*Ganesan et al., 2011*), or blasticidin-S deaminase (*Mamoun et al., 1999*) and cultured in 5 nM WR99210 (Jacobus Pharmaceutical Co.), 2 µM DSM1 (Sigma, 53330401), or 6 µM blasticidin-S (Invitrogen, R21001), respectively. After stable transfection and selection, parasites were grown in the continual presence of selection drugs, and aptamer-tagged parasites were grown in 0.5–1 µM anhydrotetrocycline (Cayman Chemicals, 10009542).

## Parasite growth assays

Parasites were synchronized to the ring stage with an estimated 10–15 hr synchrony window by treatment with 5% D-sorbitol (Sigma, S7900). For aptamer-based knockdown experiments, aTC was washed out during synchronization with additional three to five washes in media and/or PBS. Parasite growth was monitored by plating synchronized parasites at ~1% parasitemia and allowing culture expansion over several days with daily media changes. Parasitemia was monitored daily by flow cytometry by diluting 10 µL of each parasite culture well from each of three biological replicate samples into 200 µL of 1.0 µg/mL acridine orange (Invitrogen, A3568) in phosphate buffered saline (PBS) then analyzed on a BD FACSCelesta flow cytometry system monitoring SSC-A, FSC-A, PE-A, FITC-A, and PerCP-Cy5-5-A channels.

## Cloning and episomal expression of PfHO variants in parasites

The genes encoding PfHO (Pf3D7_1011900) and apicoplast ACP (Pf3D7_0208500) were PCR amplified from *P. falciparum* strain 3D7 cDNA using primers with ≥20 bp homology overhangs (primers 10–15) for ligation-independent insertion into the XhoI and AvrII sites of pTEOE (human DHFR selection cassette) in frame with a C-terminal GFP tag, and with expression driven by HSP86 promoter (*Sigala et al., 2015*). Correct plasmid sequences were confirmed by Sanger sequencing, and plasmids were transfected as described above in combination with 25 µg pHTH plasmid containing piggyBac transposase to drive stable, random integration into the parasite genome (*Balu et al., 2005*).

## Parasite genome editing to enable ligand-dependent regulation of PfHO expression

We first used restriction endonuclease-mediated integration (*Black et al., 1995*) and single-crossover homologous recombination to tag the PfHO gene to encode a C-terminal GFP-tag fused to the DHFR-destabilization domain (*Muralidharan et al., 2011*; *Armstrong and Goldberg, 2007*) and a single hemagglutinin (HA) tag in 3D7 (PM1 KO; *Liu et al., 2005*) parasites. PCR primers 18 and 19 were used to clone the 3' 1 kb DNA sequence of the PfHO gene into the XhoI and AvrII sites of the pGDB vector (*Muralidharan et al., 2011*) to serve as a homology region for integration. 50 µg of this plasmid along with 50 units of MfeI restriction enzyme (NEB R3589), which cuts at a single site within the PfHO gene just upstream of the homologous sequence cloned into the donor-repair pGDB plasmid, was transfected into 3D7 PM1 KO parasites (*Liu et al., 2005*; which express human DHFR), as described above. Parasites were positively selected with blasticidin-S in the continuous presence of trimethoprim (Sigma, T7883), cloned by limiting dilution, and genotyped by probing Southern blots of MfeI/HindIII-digested total parasite DNA with a gel-purified, 1 kb PCR product of the 3' UTR of PfHO (*Figure 3—figure supplement 1*). Southern blot signal was generated with an AlkPhos direct labeling and detection kit as previously described (*Klemba et al., 2004*; *Okada et al., 2022*). Conditional knockdown was evaluated by synchronized parasite growth assays after 3–5 x washes in PBS to remove trimethoprim.

We next used CRISPR/Cas9 and single-crossover homologous recombination to tag the PfHO gene to encode a C-terminal HA$_2$ tag fused to the glmS ribozyme (*Prommana et al., 2013*; *Winkler et al., 2004*) in Dd2 parasites. The 1 kb homology sequence or PfHO was excised from pGDB and sub-cloned by ligation into a modified pPM2GT vector (*Klemba et al., 2004*) in which the linker-GFP sequence between the AvrII and EagI sites was replaced with a HA-HA tag and stop codon followed by the 166 bp glmS ribozyme (*Prommana et al., 2013*; *Winkler et al., 2004*). QuikChange II site-directed mutagenesis (Agilent Technologies) was used with primers 20 and 21 to introduce silent shield mutations into the PfHO homology region for purposes of CRISPR/Cas9-based genome editing, such that the AGATGG sequence in the most 3' exon was changed to CGGTGG. A guide RNA sequence corresponding to TGAGTAGGAAATGGAGTAGA was cloned into a modified version of the previously published pAIO CRISPR/Cas9 vector (*Spillman et al., 2017*) in which the BtgZI site was replaced with a unique HindIII site to facilitate cloning (*Okada et al., 2022*). 50 µg each of the pPM2GT and pAIO vectors were transfected into Dd2 parasites, as described above. Parasites were positively selected by WR99210, cloned by limiting dilution, and genotyped by PCR (*Figure 3—figure supplement 1*). Conditional knockdown was evaluated by adding 0–10 mM glucosamine (Sigma, G1514) to synchronized parasite cultures and evaluating protein expression and parasite growth relative to untagged parental Dd2 parasites.

CRISPR/Cas9 and double-crossover homologous recombination was used to tag the PfHO gene to encode a single RNA aptamer at the 5' end and a 10 x aptamer cassette at the 3' end for inducible knockdown with the aptamer/TetR-DOZI system (*Nasamu et al., 2021*; *Ganesan et al., 2016*). A donor plasmid was created by ligation-independent insertion of a synthetic gene (gBlock, IDT) containing PfHO cDNA (*T. gondii* codon bias) into the linear pSN1847L vector (*Nasamu et al., 2021*), along with PCR amplified (primers 22–25) left and right homology flanks corresponding to the 5' (426 bp immediately upstream of start codon) and 3' (455 bp starting at position 47 after the TAA stop codon) untranslated regions of PfHO. Because the aptamer sequence contains two ATG motifs that can serve as alternate translation start sites, a viral 2 A peptide sequence was introduced between the 5' aptamer sequence and the start ATG of PfHO. This donor repair plasmid (50 µg) and the pAIO CRISPR/Cas9 vector (50 µg) with guide sequence TGAGTAGGAAATGGAGTAGA targeting the 3' end of the endogenous PfHO gene was transfected into Dd2 parasites, as described above. No shield mutation in the donor plasmid was required due to the altered codon bias of the synthetic PfHO cDNA in that vector. Parasites were positively selected with blasticidin-S in the presence of 1 µM aTC. Integration was confirmed by PCR and probing Southern blots of NdeI/HindIII-digested total parasite DNA with a gel-purified, 580 bp PCR product of the 3' UTR of PfHO (*Figure 3—figure supplement 4*). Southern blot signal was generated with an AlkPhos direct labeling and detection kit as previously described (*Klemba et al., 2004*; *Okada et al., 2022*). Parasites were also cloned by limiting dilution, however no evidence of remnant WT locus was detected in polyclonal transfectants and identical phenotypes were observed for polyclonal and clonal parasite cultures. Therefore, polyclonal cultures were used for all subsequent experiments.

## Microscopy

Live microscopy of parasites expressing GFP-tagged proteins was performed by staining mitochondria with 25 nM MitotrackerRed CMXRos (Invitrogen, M7512) for 30 min and staining nuclei with 5 µg/mL Hoechst 33342 (Invitrogen, H3570) for 5–10 min in PBS. Stained parasites were then imaged in PBS under a coverslip on an Invitrogen EVOS M5000. Images were adjusted for brightness and contrast in FIJI with linear variations equally applied across images. Signal intensity profiles were calculated for the red and green channels respectively using the FIJI plot profile tool along a single line that transects the region of highest signal for both channels (identified on images as white line). At least 20 individual parasites were imaged for each parasite line or each condition.

For immunofluorescence (IFA) experiments, parasitized red blood cells were fixed in 4% paraformaldehyde and 0.0016% glutaraldehyde for 30 min at 25 °C, then deposited onto poly-D-lysine coated coverslips. Fixed cells were permeabilized in PBS supplemented with 0.1% Triton-X100, reduced in 0.1 mg/mL NaBH$_4$, and blocked in 3% BSA for 30 min. Parasites were stained with primary antibodies: mouse anti-GFP (Invitrogen, A-11120), and rabbit anti-apicoplast ACP (*Gallagher and Prigge, 2010*) at 1:100 dilution in 1% BSA for 1 hr at 25 °C, washed thrice in PBS-T (PBS with 0.1% Tween-20), stained with secondary antibodies: goat anti-mouse AF488 (Invitrogen, A-11001) and goat anti-rabbit AF647 (Invitrogen, A21244) in 1% BSA for 1 hr at 25 °C, and washed thrice in PBS-T before imaging. Coverslips were mounted onto slides using ProLong Diamond Antifade Mountant with DAPI (Invitrogen, P36971) overnight at 25 °C, then imaged on an Axio Imager M1 epifluorescence microscope (Carl Zeiss Microimaging Inc) equipped with a Hamamatsu ORCA-ER digital CCD camera. Images were adjusted for brightness and contrast in FIJI with linear variations equally applied across images. Pearson correlation was calculated with the FIJI Coloc2 tool on unmasked images using a point spread function of 3 pixels and 50 Costes iterations. At least 10 individual parasites were imaged for each parasite line.

Immunogold transmission electron microscopy was performed (Dr. Wandy Beatty, Washington University in St. Louis) as previously described (*Beck et al., 2014*) using endogenously tagged PfHO-GFP-DHFR$_{DD}$ 3D7 parasites and staining with goat anti-GFP (Abcam, ab5450) and rabbit anti-apicoplast ACP (*Ponpuak et al., 2007*) antibodies along with gold-conjugated anti-goat and anti-rabbit secondary antibodies. 15 individual parasites were imaged.

## Production and validation of custom anti-PfHO rabbit antibody

The HO domain of PfHO (84-305) was cloned into pET28 with an N-terminal His-tag and start codon, purified by Ni-NTA, cleaved by thrombin, and purified by FPLC as previously reported (*Sigala et al., 2012*). Purified protein was then injected into a rabbit for polyclonal Ab production by Cocalico Biologicals Inc (https://www.cocalicobiologicals.com) following their standard protocol. Rabbit serum was validated for specific detection of PfHO in parasite lysates and with recombinant protein expressed in bacteria prior to use (*Figure 3—figure supplement 5*).

## SDS-PAGE and western blots

Parasite cultures were grown to ~10% parasitemia in 10 mL cultures for western blots and 50–100 mL cultures for immunoprecipitation. Parasites were released from red blood cells by treatment with 0.05% saponin and subsequently lysed by sonication (20 pulses 50% duty cycle, 50% power) on a Branson microtip sonicator in TBS lysis buffer (50 mM Tris pH 7.4, 150 µM NaCl, 1% v/v Triton X-100) with protease inhibitors (Invitrogen, A32955) followed by incubation at 4 °C for 1 hr. Recombinantly expressed protein was obtained for western blot analysis by inducing BL21 (DE3) *E. coli* grown to an OD of 0.5 in LB medium with 0.5 mM IPTG at 20 °C overnight. Cells were harvested by centrifugation (5000 RPM, 10 min), and lysed with the same methods described above. Lysates were clarified by centrifugation (14,000 RPM, 10 min) and quantified by Lowry colorimetry. 50 µg of total protein was mixed in SDS sample buffer, heated at 95 C for 10 min, and separated by electrophoresis on 12% SDS-PAGE gels in Tris-HCl buffer. Proteins were transferred onto nitrocellulose membranes using the Bio-Rad wet-transfer system for 1 hr at 100 V, and blocked with 5% non-fat milk in TBS-T (50 mM Tris pH 7.4, 150 µM NaCl, 0.5% v/v Tween-20). Membranes were stained with primary antibodies: goat anti-GFP (Abcam ab5450), rat anti-HA (Roche), mouse anti-hDHFR, rabbit anti-apicoplast ACP (*Gallagher and Prigge, 2010*), custom rabbit anti-PfHO, or rabbit anti-EF1α (*Mamoun and Goldberg, 2001*) diluted 1:1000-1:2500 in blocking buffer for ≥18 hr at 4 °C. Samples were then washed thrice

in TBS-T and stained with secondary antibodies: rabbit anti-Goat HRP (Santa Cruz, sc2768), goat anti-Rabbit HRP (Invitrogen, A27036), donkey anti-rabbit IRDye800CW (LiCor, 926–32213), donkey anti-rabbit IRDye680 (LiCor, 926–68023), or donkey anti-goat IRDye800CW (LiCor, 926–32214), diluted 1:10,000 in TBS-T for 1–2 hr at 25 °C, and again washed thrice before imaging. All blots stained with horseradish peroxidase (HRP) were developed 3 min with Prometheus ProSignal Femto ECL reagent (Genesee Scientific 20–302) and imaged on a BioRad ChemiDoc MP imaging system, and all blots stained with IRDye antibodies were imaged on a Licor Odyssey CLx system. Protein size was estimated by migration relative to the protein ladder using Licor Image Studio software v5.5.4.

## N-terminal protein sequencing of PfHO

A 170 mL culture of 3D7 parasites at 3% hematocrit and ~15% asynchronous parasitemia was harvested by centrifugation followed by release of parasites from RBCs in 5% saponin (Sigma S7900). The resulting parasite pellet was subsequently lysed in 1 mL RIPA buffer with sonication and clarified by centrifugation. PfHO was isolated by immunoprecipitation using 60 μL of affinity-purified custom anti-PfHO rabbit antibody and 400 μL of Protein A dynabeads (Invitrogen, 1001D). The beads were washed 3 X in RIPA buffer, eluted with 110 μL 1 X SDS sample buffer, fractionated by 10% SDS-PAGE, followed by transfer to PVDF membrane and staining by Coomassie. The band corresponding to mature PfHO was excised and subjected to N-terminal sequencing by Edman degradation at the Stanford University Protein and Nucleic Acid Facility.

## Immunoprecipitation

Dd2 parasites expressing endogenously tagged PfHO with C-terminal HA-HA tag were harvested from ~75 mL of culture by centrifugation, released from RBCs by incubating in 0.05% saponin (Sigma 84510) in PBS for 5 min at room temperature, and pelleted by centrifugation (5000 rpm, 30 min, 4 °C). Parasites were then lysed by sonication (20 pulses 50% duty cycle, 50% power) on a Branson microtip sonicator in TBS lysis buffer (50 mM Tris pH 7.4, 150 μM NaCl, 1% v/v Triton X-100) with protease inhibitors (Invitrogen A32955) followed by incubation at 4 °C for 1 hr and centrifugation (14,000 rpm, 10 min), The clarified lysates were mixed with equilibrated resin from 30 μL of Pierce anti-HA-tag magnetic beads (Invitrogen 88836) and incubated for 1 hr at 4 °C on a rotator. Beads were placed on a magnetic stand, supernatants were removed by aspiration, and beads were washed thrice with cold TBS-T. Bound proteins were eluted with 100 μLl of 8 M urea (in 100 mM Tris-HCl at pH 8.8). Proteins were precipitated by adding 100% trichloroacetic acid (Sigma 76039) to a final concentration of 20% v/v and incubated on ice for 1 hr. Proteins were then pelleted by centrifugation (13,000 rpm, 25 min, 4 °C) and washed once with 500 μL of cold acetone. The protein pellets were air-dried for 30 min and stored at −20 °C.

## Mass spectroscopy

Protein samples isolated by anti-HA-tag IP of endogenous PfHO were reduced and alkylated using 5 mM Tris (2-carboxyethyl) phosphine and 10 mM iodoacetamide, respectively, and then enzymatically digested by sequential addition of trypsin and lys-C proteases, as previously described (*Wohlschlegel, 2009*). The digested peptides were desalted using Pierce C18 tips (Thermo Fisher Scientific), dried, and resuspended in 5% formic acid. Approximately 1 μg of digested peptides was loaded onto a 25-cm-long, 75 μm inner diameter fused silica capillary packed in-house with bulk ReproSil-Pur 120 C18-AQ particles, as described previously (*Jami-Alahmadi et al., 2021*). The 140 min water-acetonitrile gradient was delivered using a Dionex Ultimate 3000 ultra-high performance liquid chromatography system (Thermo Fisher Scientific) at a flow rate of 200 nL/min (Buffer A: water with 3% DMSO and 0.1% formic acid, and Buffer B: acetonitrile with 3% DMSO and 0.1% formic acid). Eluted peptides were ionized by the application of distal 2.2 kV and introduced into the Orbitrap Fusion Lumos mass spectrometer (Thermo Fisher Scientific) and analyzed by tandem mass spectrometry. Data were acquired using a Data-Dependent Acquisition method consisting of a full MS1 scan (resolution = 120,000) followed by sequential MS2 scans (resolution = 15,000) for the remainder of the 3 s cycle time. Data was analyzed using the Integrated Proteomics Pipeline 2 (Integrated Proteomics Applications, San Diego, CA). Data were searched against the protein database from *P. falciparum* 3D7 downloaded from UniprotKB (10,826 entries) on October 2013. Tandem mass spectrometry spectra were searched using the ProLuCID algorithm followed by filtering of peptide-to-spectrum matches by DTASelect

using a decoy database-estimated false discovery rate of <1%. The proteomics data are deposited in the MassIVE data repository (https://massive.ucsd.edu) under the identifier MSV000094692.

## Measuring apicoplast DNA and RNA abundance in parasites

Highly synchronous parasites were obtained by sorbitol synchronization of high parasitemia cultures followed by magnet-purification of schizonts after 36–40 hr using MACS LD separation columns (Miltenyi, 130-042-901) with stringent washing. Purified schizonts were allowed to reinvade fresh RBCs for 5 hr on an orbital shaker at 100 rpm in media containing 1 μM aTC. Immediately prior to experimental plating, parasites were treated with sorbitol to ensure ≤5 hr synchrony window and washed three to five times in media and/or PBS to remove aTC. Times listed in growth assays are post-synchronization and reflect T=0 at the time that magnet-purified schizonts were allowed to reinvade fresh RBCs.

Highly synchronous parasites were plated in 4 mL of either +aTC or -aTC media. Parasites to be harvested in the first life cycle were cultured at 3%, second cycle at 1%, and third cycle at 0.5% starting parasitemia. 4 mL of +aTC and -aTC cultures were collected for whole DNA and RNA extraction, respectively, at 36, 84, and 132 hr, snap frozen in liquid $N_2$, and stored at –80 °C. Parasite DNA was extracted with a QIAamp DNA Blood Mini kit (QIAGEN, 51106), and RNA was purified by Trizol (Invitrogen, 15596026) and phenol-chloroform isolation. We converted 1 μg of purified RNA to cDNA using a SuperScript IV VILO RT kit (Invitrogen, 11766050). Since apicoplast genes are extremely AT- rich (*Denny et al., 1996*) and mRNA transcripts are not poly-adenylated or poly-uridylylated (*Dorrell et al., 2014*), gene-specific reverse primers were used to prime the reverse transcription reactions. RT-qPCR was then used to assess the DNA/RNA abundance of four nuclear genes: STL (Pf3D7_0717700), I5P (Pf3D7_0802500), ADSL (Pf3D7_0206700), and PfHO (Pf3D7_1011900), one mitochondrial gene: CytB (Pf3D7_MIT02300), and twelve apicoplast genes: rpl-4 (Pf3D7_API01300), rpl-2 (Pf3D7_API01500), rpl-14 (Pf3D7_API02000), rps-12 (Pf3D7_API02700), EF-Tu (Pf3D7_API02900), ClpM (Pf3D7_API03600), RpoC2 (Pf3D7_API04200), RpoC1 (Pf3D7_API04300), RpoB (Pf3D7_API04400), SufB (Pf3D7_API04700), ls-rRNA (Pf3D7_API06700), and ss-rRNA (Pf3D7_API05700) (primers 28–61). Invitrogen Quantstudio Real-Time PCR systems were used to quantify abundance of DNA and cDNA using SYBR green dye and primers 28–61. The relative DNA or cDNA abundance of each apicoplast gene was normalized to the average of three nuclear-encoded genes for each sample, and -aTC was compared to +aTC by the comparative Ct method (*Schmittgen and Livak, 2008*). All qPCR experiments were performed in triplicate and data was analyzed by unpaired Student's t-test.

## PfHO chromatin immunoprecipitation (ChIP) analysis

We saponin-released 75 mL of high parasitemia Dd2 cultures transfected with episomes encoding expression of PfHO-GFP, PfHO[1-83]-GFP, ACP$_L$-PfHO[84-305]-GFP, and ACP$_L$-GFP and crosslinked in 1% paraformaldehyde for 15 min at 20 °C, then quenched with 125 mM glycine. Crosslinked parasites were transferred into 2 mL ChIP lysis/sonication buffer (200 mM NaCl, 25 mM Tris pH 7.5, 5 mM EDTA pH 8, 1% v/v Triton X-100, 0.1% SDS w/v, 0.5% sodium deoxycholate w/v, and protease inhibitors), and sonicated for 15 cycles of 30 s ON/OFF at 25% power using a microtip on a Branson sonicator, then clarified by centrifugation (14,000 rpm, 10 min, 4 °C). DNA fragment size after shearing was determined by Agilent Bioanalyzer DNA analysis (University of Utah DNA Sequencing Core; *Figure 5— figure supplement 4*). We collected 200 μL (10%) of the clarified, sheared lysates as 'input controls' and the incubated the rest with goat anti-GFP antibody (Abcam, ab5450) at 4 °C overnight. Antibody-bound protein was mixed with equilibrated resin from 25 μL protein A-conjugated dynabeads (Invitrogen, 1011D) for 1 hr at 4 °C rotating, then washed in lysis/sonication buffer, 1 mg/mL salmon sperm DNA (Invitrogen, AM9680) in lysis/sonication buffer, high-salt wash buffer (500 mM NaCl, 25 mM Tris pH 7.5, 2 mM EDTA pH 8, 1% v/v Triton X-100, 0.1% w/v SDS, and protease inhibitors), and Tris-EDTA buffer. Samples were eluted from protein A dynabeads by two rounds of 5 min incubation at 65 °C in 100 μL elution buffer (10 mM Tris pH 8, 1 mM EDTA, 1% w/v SDS). We increased NaCl concentration in both input control and ChIP elution samples to 200 mM and added 50 μg/mL RNAse A, then incubated overnight (at least 8 hr) at 65 °C to reverse crosslinks and digest RNA. We increased EDTA concentration to 5 mM and added 2 μL of 20 mg/mL Proteinase K and digested at 60 °C for 1 hr, then purified DNA using the Qiagen PCR purification kit. Purified DNA was immediately used for either steady-state PCR using primers 62–73 or qPCR amplification using protocol described above.

Relative quantification of steady-state PCR bands was performed by area-under-the-curve densitometry analysis in FIJI. In qPCR experiments, amplification of each gene in ChIP DNA was normalized to amplification of the same gene in DNA purified from the input control to account for variability between parasite lines. All densitometry and qPCR experiments were performed in triplicate and statistical significance of differences between PfHO-GFP and other constructs was calculated using Student's t-test.

## Materials availability

All materials created during this study can be obtained by contacting the Sigala lab.

## Acknowledgements

We thank Wandy Beatty and Josh Beck for assistance with electron microscopy and western blot experiments, respectively, and thank Geoff McFadden, Jacquin Niles, Akhil Vaidya, Dennis Winge, and members of the Goldberg and Sigala labs for helpful discussions. We thank Dick Winant at the Stanford PAN facility for assistance with N-terminal protein sequencing. We thank the staff of the Photon factory for excellent technical support. Access to beamline BL5A and AR-NW12A was granted by the Photon Factory Advisory Committee (Proposals 2011G574, and 2012 G191). PAS holds a Burroughs Wellcome Fund Career Award at the Scientific Interface and is a Pew Biomedical Scholar, supported by the Pew Charitable Trusts. DNA synthesis and sequencing, fluorescence microscopy, and flow cytometry were performed using core facilities at the University of Utah.

## Additional information

### Funding

| Funder | Grant reference number | Author |
|---|---|---|
| National Institute of General Medical Sciences | R35GM153408 | James A Wohlschlegel |
| National Institute of General Medical Sciences | R35GM133764 | Paul A Sigala |
| National Institute of Allergy and Infectious Diseases | R21AI110712 | Daniel E Goldberg |
| National Institute of Diabetes and Digestive and Kidney Diseases | T32DK007115 | Amanda Mixon Blackwell |
| Burroughs Wellcome Fund | 1011969 | Paul A Sigala |
| Pew Charitable Trusts | 32099 | Paul A Sigala |
| National Human Genome Research Institute | R25HG009886 | Celine Slam |

The funders had no role in study design, data collection and interpretation, or the decision to submit the work for publication.

### Author contributions

Amanda Mixon Blackwell, Formal analysis, Investigation, Visualization, Writing - original draft, Writing - review and editing; Yasaman Jami-Alahmadi, Armiyaw S Nasamu, Shota Kudo, Akinobu Senoo, Investigation; Celine Slam, Formal analysis, Investigation; Kouhei Tsumoto, Funding acquisition, Project administration; James A Wohlschlegel, Supervision, Funding acquisition, Validation; Jose Manuel Martinez Caaveiro, Data curation, Formal analysis, Validation, Investigation, Project administration; Daniel E Goldberg, Conceptualization, Supervision, Funding acquisition, Project administration; Paul A Sigala, Conceptualization, Formal analysis, Supervision, Funding acquisition, Investigation, Visualization, Methodology, Project administration, Writing - review and editing

## Author ORCIDs

Amanda Mixon Blackwell (iD) https://orcid.org/0000-0002-7473-5822
Yasaman Jami-Alahmadi (iD) https://orcid.org/0000-0001-8289-2222
Daniel E Goldberg (iD) https://orcid.org/0000-0003-3529-8399
Paul A Sigala (iD) https://orcid.org/0000-0002-3464-3042

Reviewer #1 (Public review): https://doi.org/10.7554/eLife.100256.3.sa1
Reviewer #2 (Public review): https://doi.org/10.7554/eLife.100256.3.sa2
Author response https://doi.org/10.7554/eLife.100256.3.sa3

---

# Additional files

## Supplementary files

- Supplementary file 1. Table of PCR primers.
- MDAR checklist

## Data availability

Atomic coordinates and structure factors for PfHO have been deposited in the RCSB Protein Data Bank as entry 8ZLD. The proteomics data are deposited in the MassIVE data repository (https://massive.ucsd.edu) under the identifier MSV000094692. All data generated or analyzed during this study are included in the manuscript and supporting files; source data files have been provided for **Figures 1–5**. **Figure 5—source data 1** contains the list of proteins identified in PfHO IP/MS experiments. **Supplementary file 1** contains a list of PCR primers and sequences.

The following datasets were generated:

| Author(s) | Year | Dataset title | Dataset URL | Database and Identifier |
|---|---|---|---|---|
| Caaveiro JMM, Senoo A | 2024 | Crystal structure of PfHO from *Plasmodium falciparum* at 2.78 A | https://www.rcsb.org/structure/8ZLD | RCSB Protein Data Bank, 8ZLD |
| Wohlschlegel JA | 2024 | Malaria parasites require a divergent heme oxygenase for apicoplast gene expression and biogenesis | https://doi.org/10.25345/C50K26P12 | MassIVE, 10.25345/C50K26P12 |

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
