## [Editor Report · eLife Assessment]

This study reveals that the malaria parasite protein PfHO, although lacking typical heme oxygenase activity, is essential for the survival of *Plasmodium falciparum*. Structural and localization analyses demonstrated that PfHO plays a critical role in maintaining the apicoplast, specifically in gene expression and biogenesis, suggesting an adaptive function for this protein in parasite biology. While the findings **convincingly** support the authors' claims, further investigation into apicoplast gene expression and the specific function of PfHO remains a future challenge. The topic and results are **important** and will be of interest to researchers studying various aspects of malaria, Plasmodium physiology, host-pathogen interactions, and heme metabolism.

---

## [Referee Report · Reviewer #1 (Public review)]

Malaria parasites detoxify free heme molecules released from digested host hemoglobins by biomineralizing them into inert hemozoin. Thus, why malaria parasites retain PfHO, a dead enzyme that loses the capacity of catabolizing heme, is an outstanding question that has puzzled researchers for more than a decade. In the current manuscript, the authors addressed this question by first solving the crystal structure of PfHO and aligning it with structures of other heme oxygenase (HO) proteins. They found that the N-terminal 95 residues of PfHO, which failed to crystalize due to its disordered nature, may serve as signal and transit peptides for PfHO subcellular localization. This was confirmed by subsequent microscopic analysis with episomally expressed PfHO-GFP and a GFP reporter fused to the first 83 residues of PfHO (PfHO N-term-GFP). To investigate the functional importance of PfHO, the authors generated an anhydrotetracycline (aTC) controlled PfHO knockdown strain. Strikingly, the parasites lacking PfHO failed to grow and lost their apicoplast. Finally, by chromatin immunoprecipitation (ChIP), quantitative PCR/RT-PCR and growth assays, the authors showed that both the cognate N-terminus and HO-like domain were required for PfHO function as an apicoplast DNA interacting protein.

The authors systemically performed multidisciplinary approaches to address this difficult question: what is the function of this enzymatically dead PfHO? I enjoyed reading this manuscript and its thoughtful discussion. This study is not only of clinical importance for antimalarial treatments but also deepens our understanding of protein function evolution.

The authors proposed that PfHO interacts with apicoplast genome DNA via the electropositive N-terminus. Interestingly, these positively charged residues are not conserved between Plasmodium, Theileria and Babesia. I will be curious to follow the authors' future work to investigate the function of this electropositive N-terminus, possibly by comparative and mutagenesis analysis?

---

## [Referee Report · Reviewer #2 (Public review)]

Summary:

Blackwell et al. investigated the structure, localization and physiological function of *Plasmodium falciparum* (Pf) heme oxygenase (HO). Pf and other malaria parasites scavenge and digest large amounts of hemoglobin from red cells for sustenance. To counter the potentially cytotoxic effects of heme, it is biomineralized into hemozoin and stored in the food vacuole. Another mechanism to counteract heme toxicity is through its enzymatic degradation via heme oxygenases. However, it was previously found by the authors that PfHO lacks the ability to catalyze heme degradation, raising the intriguing question of what the physiological function of PfHO is. In the current contribution, the authors determine that PfHO localizes to the apicoplast, determine its targeting sequence, establish the essentiality of PfHO for parasite viability, and determine that PfHO is required for proper maintenance of apicoplasts and apicoplast gene expression. In sum, the authors establish an essential physiological function for PfHO, thereby providing new insights into the role of PfHO in plasmodium metabolism.

Strengths:

The studies are rigorously conducted and the results of the experiments unambiguously support a role for PfHO as being an apicoplast targeted protein required for parasite viability and maintenance of apicoplasts.

Weaknesses:

While the studies conducted are rigorous and support the primary conclusions, the lack of experiments probing the molecular function of PfHO somewhat limits the impact of the work. Nevertheless, knowledge that PfHO is required for parasite viability and plays a role in the maintenance of apicoplasts is still an important advance.

Comments on revisions:

The authors thoughtfully addressed all the reviewer comments.

---

## [Author Response]

The following is the authors’ response to the original reviews.

**eLife Assessment:**
This important study reveals that the malaria parasite protein PfHO, though lacking typical heme oxygenase activity, is vital for the survival of *Plasmodium falciparum*. Structural and localization analyses showed that PfHO is essential for apicoplast maintenance, particularly in gene expression and biogenesis, indicating a novel adaptive role for this protein in parasite biology. While the results supporting the claims of the authors are convincing, the lack of data defining a molecular understanding or mechanism of action of the protein in question limits the impact of the study.

We appreciate the positive assessment. We agree that further mechanistic understanding of PfHO function remains a key future challenge. Indeed, we made extensive efforts to unravel the molecular interactions and mechanisms that underpin the critical function of PfHO. We elucidated key interactions between PfHO and the apicoplast genome, reliance of these interactions on the electropositive N-terminus, association of PfHO with DNA-binding proteins, and a specific defect in apicoplast mRNA levels upon PfHO knockdown. The major limitation we faced in further defining PfHO function is the general lack of understanding of apicoplast transcription and broader gene expression in this organelle. That limitation and the challenges to overcome it go well beyond our study and will require concerted efforts across several manuscripts (likely by multiple groups) to define the mechanistic features of apicoplast gene expression. We look forward to contributing further molecular understanding of PfHO function as broader understanding of apicoplast transcription emerges.

**Public Reviews:**

**Reviewer #1 (Public Review):**
Malaria parasites detoxify free heme molecules released from digested host hemoglobins by biomineralizing them into inert hemozoin. Thus, why malaria parasites retain PfHO, a dead enzyme that loses the capacity of catabolizing heme, is an outstanding question that has puzzled researchers for more than a decade. In the current manuscript, the authors addressed this question by first solving the crystal structure of PfHO and aligning it with structures of other heme oxygenase (HO) proteins. They found that the N-terminal 95 residues of PfHO, which failed to crystalize due to their disordered nature, may serve as signal and transit peptides for PfHO subcellular localization. This was confirmed by subsequent microscopic analysis with episomally expressed PfHO-GFP and a GFP reporter fused to the first 83 residues of PfHO (PfHO N-term-GFP). To investigate the functional importance of PfHO, the authors generated an anhydrotetracycline (aTC) controlled PfHO knockdown strain. Strikingly, the parasites lacking PfHO failed to grow and lost their apicoplast. Finally, by chromatin immunoprecipitation (ChIP), quantitative PCR/RT-PCR, and growth assays, the authors showed that both the cognate N-terminus and HO-like domain were required for PfHO function as an apicoplast DNA interacting protein.The authors systemically performed multidisciplinary approaches to address this difficult question: what is the function of this enzymatically dead PfHO? I enjoyed reading this manuscript and its thoughtful discussion. This study is not of clinical importance for antimalarial treatments but also deepens our understanding of protein function evolution. While I understand these experiments are challenging to conduct in malaria parasites, the data quality of some of the experiments could be improved. For example, most of the Western blots and Southern blots are not of high quality.

We thank the reviewer for the positive comments but are a bit puzzled by the final statement about western and Southern blot quality. We agree that the two anti-PfHO western blots probed with custom antibody (Fig. 3- source data 2 and 8) have substantial background signal in the higher molecular mass region >75 kDa. However, we note that the critical region <50 kDa is clear in both cases and readily enables target band visualization. All other western blots probing GFP or HA epitopes are of high quality with minimal off-target background. We present two Southern blot images. We agree that the signal is somewhat faint for the Southern blot demonstrating on-target integration of the aptamer/TetR-DOZI plasmid (Fig. 3- fig. supplement 4), although we note that the correct band pattern for integration is visible. We also note that the accompanying genomic PCR data is unambiguous. The Southern blot for GFPDHFRDD incorporation into the PfHO locus (Fig. 3- fig. supplement 1) has clear signal and strongly supports on-target integration. The minor background signal in the lower left region of the image does not extend into the critical lanes nor impact interpretation of correct clonal integration.

As noted below, we have obtained a second western blot image to evaluate the decrease in PfHO protein expression in -aTC conditions. This revised image, which we now include in Fig. 3, shows clean detection of the PfHO signal in the critical molecular mass region below 40 kDa in +aTC conditions and substantial loss of this signal in -aTC conditions (relative to HSP60 loading control).

**Reviewer #2 (Public Review):**
Summary:Blackwell et al. investigated the structure, localization, and physiological function of *Plasmodium falciparum* (Pf) heme oxygenase (HO). Pf and other malaria parasites scavenge and digest large amounts of hemoglobin from red cells for sustenance. To counter the potentially cytotoxic effects of heme, it is biomineralized into hemozoin and stored in the food vacuole. Another mechanism to counteract heme toxicity is through its enzymatic degradation via heme oxygenases. However, it was previously found by the authors that PfHO lacks the ability to catalyze heme degradation, raising the intriguing question of what the physiological function of PfHO is. In the current contribution, the authors determine that PfHO localizes to the apicoplast, determine its targeting sequence, establish the essentiality of PfHO for parasite viability, and determine that PfHO is required for proper maintenance of apicoplasts and apicoplast gene expression. In sum, the authors establish an essential physiological function for PfHO, thereby providing new insights into the role of PfHO in plasmodium metabolism.Strengths:The studies are rigorously conducted and the results of the experiments unambiguously support a role for PfHO as being an apicoplast-targeted protein required for parasite viability and maintenance of apicoplasts.Weaknesses:While the studies conducted are rigorous and support the primary conclusions, the lack of experiments probing the molecular function of PfHO limits the impact of the work. Nevertheless, the knowledge that PfHO is required for parasite viability and plays a role in the maintenance of apicoplasts is still an important advance.

We appreciate the positive assessment. We agree that further mechanistic understanding of PfHO function remains a key future challenge. Indeed, we made extensive efforts to unravel the molecular interactions and mechanisms that underpin the critical function of PfHO. We elucidated key interactions between PfHO and the apicoplast genome, reliance of these interactions on the electropositive N-terminus, association of PfHO with DNA-binding proteins, and a specific defect in apicoplast mRNA levels upon PfHO knockdown. The major limitation we faced in further defining PfHO function is the general lack of understanding of apicoplast transcription and broader gene expression. That limitation and the challenges to overcome it go well beyond our study and will require concerted efforts across several manuscripts (likely by multiple groups) to define the mechanistic features of apicoplast gene expression. We look forward to contributing further molecular understanding of PfHO function as broader understanding of apicoplast transcription emerges.

**Recommendations for the authors:**

**Reviewer #1 (Recommendations For The Authors):**
Specifically, I would like to see the expression of PfHO in the 3D7 strain and PfHOaptamer/TetR-DOZI parasites detected by PfHO antibody on the same blot. The reason is that while most of the western blots show that PfHO appears as both pro- and processed-form, Figure 3-S5B shows only the processed-form of PfHO in all life stages of 3D7. It would be interesting to find out if the processing of PfHO1 is strain/stage-specific, and whether it is regulated by heme levels. It may also be interesting to find out if the pro-form of PfHO is also functional (i.e. mutate the cleavage site).

We agree with the reviewer that Fig. 3- figure supplement 5B shows predominant detection of a single band for PfHO in untagged 3D7 parasites. In our experience, the detection of the unprocessed, pro form of PfHO can vary idiosyncratically with different experiments and cultures. In support of this variable detection of unprocessed PfHO in 3D7, we note in Fig. 3A that we detected both the unprocessed and processed forms of PfHO in a western blot of endogenously tagged PfHO-GFP-DHFRDD in 3D7 parasites with an intact apicoplast. We agree with the reviewer that future studies of stage-dependent processing of PfHO may give insights into conditions that favor or disfavor detection of the unprocessed protein.

Given prior evidence for vestigial heme binding by PfHO (Sigala et al. JBC 2012), we considered whether such heme binding might modulate PfHO expression, stability, and/or function. It is unknown if heme is present inside the apicoplast, and we currently lack evidence for heme-dependent function or expression by PfHO. Future studies can test this possible dependence.

Regarding processing and possible function of the cleaved peptide, we note that the Nterminal 18 amino acids are expected to constitute the signal peptide that is cleaved cotranslationally with import into the ER. Our data indicate that PfHO undergoes further processing upon import into the apicoplast to remove a further 15 residues. We currently have no evidence nor expectation that these additional residues contribute to PfHO function beyond targeting to the apicoplast.

I am also confused as to why the authors used rabbit anti-PfHO and rabbit anti-Ef1α on the same blot for Figure 3C, which makes it difficult to appreciate the expression changes of PfHO. Given the high non-specific background of PfHO antibody shown by other Western blots (Figure 3 - Source data 2), I would like to see a blot stained with only PfHO antibody to show that expression of PfHO has been efficiently reduced in the absence of aTC.

Bands for Ef1α (50 kDa) and untagged PfHO (~32 kDa) are readily distinguished by western blot analysis based on their distinct molecular masses and electrophoretic mobilities. We agree that staining with the anti-PfHO antibody resulted in background bands in other regions of the gel image, especially in the higher molecular mass region >75 kDa. We note that additional strong evidence for down-regulation of PfHO expression is provided in Fig. 3- figure supplement 6, which shows specific loss of PfHO mRNA transcript levels in -aTC conditions by RT-qPCR.

Nevertheless, we have followed the reviewer’s suggestion and provided a new WB image of PfHO expression ±aTC (probed only with rabbit anti-PfHO antibody) that shows strong down-regulation of PfHO protein levels in -aTC conditions, consistent with the strong growth phenotype observed. We have inserted this revised, cleaner western blot image into Fig. 3 (along with detection of HSP60 levels in replicate samples as loading control) and placed the prior image into Fig. 3- figure supplement 6. In both cases, densitometry analysis indicates an 80-85% reduction in PfHO levels in -aTC conditions.

The authors proposed that PfHO interacts with apicoplast genome DNA via the electropositive N-terminus. Interestingly, these positively charged residues are not conserved between Plasmodium, Theileria, and Babesia. I will be curious to follow the authors' future work to investigate the function of this electropositive N-terminus, possibly by comparative and mutagenesis analysis.

We agree that further molecular studies of DNA-binding determinants by PfHO and its N-terminus will be insightful.

The Quantitative RT-PCR analysis revealed that loss of PfHO specifically resulted in decreased apicoplast RNA. I wonder if the authors plan to conduct RNAseq analysis on the PfHO knockdown strain across multiple life stages, to get a clearer picture of PfHO function in malaria parasites.

Our RT-qPCR data across multiple asexual stages prior to organelle loss indicate that abundance of all apicoplast-encoded transcripts drops precipitously and uniformly upon PfHO knockdown (Fig. 5- figure supplement 7). Given the small size of the apicoplast genome and the polycistronic nature of apicoplast transcription, we assume that RNA-Seq studies would result in a similar observation. We hypothesize that PfHO knockdown and subsequent dysfunctions may interfere with RNA polymerase assembly on DNA and/or processivity. We are currently testing these hypotheses.

I noticed that the authors did not discuss the function of PfHO in apicoplast organelle biogenesis. Since ClpM (previously termed ClpC) is the only apicoplast-encoded Clp subunit that is essential for apicoplast biogenesis, does the author think that PfHO knockdown parasites lost their apicoplast due to decreased ClpM expression? If that were the case, would episomally expression or nuclear knockin of ClpM rescue PfHO deficiency in the absence of isopentenyl pyrophosphate (IPP)?

We share the reviewer’s curiosity to understand how loss of apicoplast transcripts leads to organelle dysfunction and defective IPP synthesis. We agree that ClpM function may be critical to import of nuclear-encoded proteins necessary for apicoplast function. SufB encoded on the apicoplast genome is also expected to be essential for Fe-S cluster synthesis in the apicoplast and to be required for Fe-S-dependent IPP synthesis. We have expanded the first Discussion section to address these possible connections.

Minor:(1) None of the microscopy photos have scale bars.

We have added scale bars to all microscopy images.

(2) Multiple microscopy pictures show strange patches around the fluorescent signals (a grey square distinguishes from the black background). This is especially evident in Figure 2 S2. Was it caused by the reduction of the original pictures?

We have reviewed all fluorescence microscopy images but are unable to identify the issue noted by the reviewer. We have uploaded new versions of all images to include scale bars (as requested above), and we hope that this update resolves the issue observed by the reviewer. We are happy to further troubleshoot and address if the reviewer continues to see these artifacts and can provide further information.

(3) A description of how Southern blotting was performed is missing.

We thank the reviewer for bringing this omission to our attention. We have added a description of the Southern blot methods to the section on genome editing.

(4) Figure 3B: should be "αGFP: 12nm", not "αPfHO1: 12nm".

We have modified this labeling to read “αGFP (PfHO): 12 nm”.

(5) Figure 3C: which clone of PfHO knockdown was used in all the following figures? How many clones were tested in the following figures (did they show consistent phenotype)?

The polyclonal culture of PfHO-aptamer/TetR-DOZI knockdown parasites from transfection 11 was used for growth assay and western blot experiments, since there was no evidence by PCR or Southern blot for the wildtype PfHO locus. We have elaborated on these details in the Methods section.

**Reviewer #2 (Recommendations For The Authors):**
In Figure 2 and Figure 3B, to address rigor and reproducibility, the authors should state the number of parasites analyzed and if there was any variation in localization. For instance, did all of the parasites analyzed have apicoplast localization of heme oxygenase or was there a distribution of apicoplast and non-apicoplast localization?

Localization by fluorescence microscopy of episomal and endogenous tagged PfHO is presented in Fig. 2, Fig. 2- fig. supplements 1 and 2, and Fig. 3- fig. supplement 2. Localization by immunogold EM is presented in Fig. 3B and Fig. 3- fig. supplement 3. In all cases 3-4 representative images are presented that support exclusive localization of PfHO to the apicoplast. We imaged ≥10-20 additional parasites in all cases (and across distinct transfections and biological samples) that also supported exclusive localization to the apicoplast. We have modified the figure legends and methods description to note these replicate values. Finally, we note that IPP rescue of parasite viability upon PfHO knockdown strongly supports the conclusion that the critical and essential function of PfHO impacts the apicoplast, consistent with its exclusive detection in that organelle by microscopy.